# Structural asymmetry governs the assembly and GTPase activity of McrBC restriction complexes

Yiming Niu [1,2,3,6], Hiroshi Suzuki [2,4,6], Christopher J. Hosford[1,5], Thomas Walz [2✉] &
Joshua S. Chappie [1✉]

McrBC complexes are motor-driven nucleases functioning in bacterial self-defense by cleaving foreign DNA. The GTP-specific AAA + protein McrB powers translocation along DNA and its hydrolysis activity is stimulated by its partner nuclease McrC. Here, we report cryo-EM structures of *Thermococcus gammatolerans* McrB and McrBC, and *E. coli* McrBC. The McrB hexamers, containing the necessary catalytic machinery for basal GTP hydrolysis, are intrinsically asymmetric. This asymmetry directs McrC binding so that it engages a single active site, where it then uses an arginine/lysine-mediated hydrogen-bonding network to reposition the asparagine in the McrB signature motif for optimal catalytic function. While the two McrBC complexes use different DNA-binding domains, these contribute to the same general GTP-recognition mechanism employed by all G proteins. Asymmetry also induces distinct inter-subunit interactions around the ring, suggesting a coordinated and directional GTP-hydrolysis cycle. Our data provide insights into the conserved molecular mechanisms governing McrB family AAA + motors.

[1] Department of Molecular Medicine, Cornell University, Ithaca, NY, USA. [2] Laboratory of Molecular Electron Microscopy, The Rockefeller University, New York, NY, USA. [3] Present address: Laboratory Molecular Neurobiology and Biophysics, The Rockefeller University, New York, NY, USA. [4] Present address: Advanced Research Institute, Tokyo Medical and Dental University, Tokyo, Japan. [5] Present address: New England Biolabs, Inc., Ipswich, MA, USA. [6] These authors contributed equally: Yiming Niu, Hiroshi Suzuki. ✉email: twalz@rockefeller.edu; chappie@cornell.edu

Infections by antibiotic-resistant bacteria pose a serious threat to human health[1,2]. The slow progress in developing new drugs to combat these emerging "superbugs" and the rapid exchange of resistance genes among microbial populations has intensified the need for alternative therapeutic strategies[3]. One such strategy employs bacteriophages (phages)—viruses that infect a bacterial host, replicate, and then induce cell lysis to release the mature phage progeny, killing the host in the process[4]. The pharmaceutical application of phages dates back to the early 1920s[5] and has resurged in recent years, bolstered by success in a number of clinical settings[6,7]. Despite these promising results, phage therapy faces numerous challenges. One significant hurdle is that bacteria have evolved an array of defense mechanisms, including restriction modification systems, modification-dependent restrictions systems (MDRSs), phage-exclusion systems, and CRISPR-Cas adaptive immune systems, that can hinder phage infection and diminish their subsequent killing potential[8,9]. These machineries lack eukaryotic homologs and are conserved across antibiotic-resistant bacteria like methicillin-resistant *Staphylococcus aureus*, *Clostridium difficile*, and *Klebsiella pneumoniae*, making their components promising candidates for targeted inhibition. Some phages indeed already encode inhibitor proteins that can neutralize restriction and/or CRISPR systems[10,11], allowing them to survive and kill under conditions in which they would normally be suppressed. Elucidating the structure and function of bacterial defense systems will therefore extend these principles and aid in the development of new drugs that increase phage efficacy.

McrBC is a two-component MDRS that in *Escherichia coli* (Ec) restricts phage DNA and foreign DNA containing methylated cytosines[12,13]. EcMcrB consists of an N-terminal DNA-binding domain that targets fully or hemi-methylated R$^M$C sites (where R is a purine base and $^M$C is a 4-methyl-, 5-methyl-, or 5-hydroxymethyl-cytosine)[14–19], and a C-terminal AAA+ (extended ATPases Associated with various cellular Activities) domain that hydrolyzes GTP and oligomerizes into hexamers[20,21]. EcMcrB's basal GTPase activity (~0.5–1 min$^{-1}$) is stimulated ~30–40-fold in vitro via interaction with its partner EcMcrC[16], a PD-(D/E)xK family endonuclease that cannot stably bind DNA on its own, and thus associates with the hexameric McrB AAA+ ring[21]. Biochemical data suggest that stimulated GTP hydrolysis powers DNA translocation[18,22], allowing EcMcrBC complexes bound to distant R$^M$C sites to interact and induce cleavage on both strands[23,24]. While these activities have yet to be demonstrated in vitro for homologs beyond *E. coli*, other family members have also been shown to function in bacterial defense in vivo[25–27]. These machines, however, exhibit different specificities for DNA modifications and/or sequences[25,27–30], suggesting that the core machinery for GTP hydrolysis and DNA cleavage is conserved and has been adapted to different targets throughout evolution in response to various selective pressures from invading phages. This flexibility holds a tremendous potential for engineering new endonucleases for biotechnology and biomedical applications, providing further motivation to study the structural organization and functional regulation of McrBC complexes.

AAA+ proteins are large, multimeric machines that use the energy of ATP hydrolysis to power a wide array of cellular processes[31]. These enzymes are built around a common structural core[32] and contain numerous conserved sequence elements important for nucleotide binding and hydrolysis[33]. AAA+ protein active sites are formed at the interface between two monomers, thus requiring higher-order assembly—predominantly as hexamers—for function[34]. As a consequence, some catalytic residues like charge-compensating arginine fingers are provided in *trans* by the neighboring subunit. Despite sharing a common architecture, McrB is the only AAA+ protein that preferentially binds and hydrolyzes GTP[35,36]. All McrB homologs contain a conserved consensus sequence of MNxxDRS that replaces the AAA+ sensor I motif and is predicted to function as a G4 element, which confers guanine nucleotide specificity in GTPases[37]. Mutation of this segment, however, does not significantly alter the nucleotide-binding profile of *E. coli* McrB[16,36], indicating that other regions of the protein dictate GTP selectivity. Stimulation of hydrolysis by a binding partner is rare among AAA+ proteins[38,39], but reminiscent of the activation of small GTPases by their corresponding GTPase-activating proteins (GAPs)[40]. A key difference from other GTPases in this instance, however, is that the second component McrC only exerts its effects on the assembled McrB oligomer. Elucidating the structural basis for GTP recognition and stimulated hydrolysis is important for defining McrBC's divergence from other members of both the AAA+ and GTPase superfamilies.

A recent cryo-EM reconstruction of the hexameric EcMcrB AAA+ domain bound to EcMcrC at 3.6-Å; resolution[41] provided the first glimpse of this machine, showing the overall architecture of the complex and proposing a general mechanism for catalytic turnover. However, this study did not resolve the molecular details and chemistry underlying the GTP hydrolysis reaction and its stimulation by McrC and may not have identified the correct DNA-binding mode. Here, we present cryo-EM structures of an McrB hexamer and McrBC complexes from the evolutionarily distant archaeal species *Thermococcus gammatolerans* (Tg) and the well-characterized *E. coli* system. Our models confirm that McrBC complexes share the same general architecture, but lead to a different view of the GTP hydrolysis cycle wherein structural asymmetry drives the underlying physical interactions and conformational motions. Moreover, our structures provide a detailed molecular mechanism for how McrC-binding stimulates McrB GTP hydrolysis, which we show is conserved across the McrBC family. Our structures also establish that McrB homologs use the same general chemistry employed by all GTPases to recognize GTP, albeit through different structural elements upstream of the AAA+ domain. This observation establishes how distant McrB homologs have adapted and maintained guanine nucleotide specificity despite the individual constraints imposed by their structurally unrelated N-terminal domains. Together these data provide mechanistic insights into the structure, function, and regulation of motor-driven McrBC nucleases.

## Results

**TgMcrB$^{AAA}$ forms an asymmetric hexamer.** Given the widespread distribution of *mcrBC* genes among diverse bacteria and archaea, we sought to examine the structural and biochemical properties of different McrB homologs to understand how these AAA+ enzymes have evolved to preferentially bind and hydrolyze GTP. Our previous work identified the archaeal McrB homolog from *T. gammatolerans* (TgMcrB) as an ideal candidate for structural studies given its compact size and increased thermostability[29]. The purified AAA+ domain from TgMcrB (TgMcrB$^{AAA}$) forms stable oligomers even in the absence of nucleotides (Supplementary Fig. 1a). Single-particle cryo-EM analysis of purified TgMcrB$^{AAA}$ incubated with the non-hydrolyzable GTP analog GTPγS yielded a density map at an overall resolution of 3.1 Å with no symmetry imposed (Fig. 1 and Supplementary Fig. 1). The cryo-EM map reveals that TgMcrB$^{AAA}$ forms a ring-shaped, homohexameric assembly with six nucleotides bound at the subunit interfaces, similar to the closed-ring assembly seen in type I AAA ATPases[42,43]. Each subunit displays a canonical AAA+ fold with the additional features of a β-hairpin inserted in helix 2 of the large subdomain

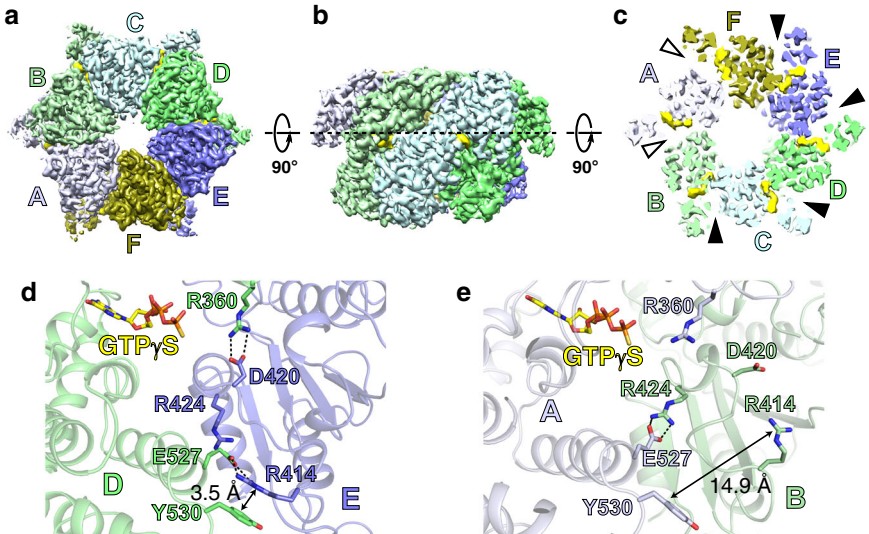

**Fig. 1 TgMcrB^AAA forms an asymmetric hexamer. a, b** Bottom and side views of the cryo-EM density map of the TgMcrB^AAA hexamer. Subunits are colored in shades of blue and green, and nucleotides are shown in yellow. **c** Slice section through the TgMcrB^AAA hexamer map at the level of the bound nucleotides, indicated by the dashed line in **b**. Solid and empty arrowheads indicate tight and loose interfaces, respectively. **d, e** Close-up views of interacting residues at the tight D/E interface (**d**) and the loose A/B interface (**e**). Dashed lines indicate hydrogen bonds.

as previously predicted[44] and "wing"-like helices in the small subdomain (Supplementary Figs. 2 and 3). The TgMcrB^AAA hexamer is asymmetric with four tight interfaces (between monomers B/C, C/D, D/E, and E/F) that bury a surface area ranging from 2393 to 2554 Å$^2$, and two loose interfaces (between monomers A/B and F/A) that bury surface areas of 1519 and 1772 Å$^2$ (Fig. 1c). Tight interfaces feature a hydrogen bond between Asp420 in one monomer and Arg360 in the adjacent monomer (Fig. 1d). Arg414 from the first monomer also extends into the neighboring monomer, where it forms hydrogen bonds with Glu527 and π-stacking interactions with Tyr530 (Fig. 1d). These interactions are absent at the loose interfaces, where Glu527 instead interacts in *trans* with Arg424 (Fig. 1e). All of these residues are highly conserved amongst McrB family proteins (Supplementary Fig. 2).

To determine if these interface residues affect McrB's catalytic turnover, we mutated each side chain individually to alanine in the context of TgMcrB^AAA and measured basal GTPase activity using a colorimetric assay. All mutants show an approximate twofold increase in hydrolysis activity compared to that of the wild-type protein (Supplementary Fig. 1i). Alanine substitution of Arg337 in EcMcrB (corresponding to Arg414 in TgMcrB, Supplementary Fig. 3) was previously shown to increase the basal GTPase rate threefold[16], consistent with our results.

In parallel, we also determined the structure of TgMcrB^AAA in the presence of GTPγS by X-ray crystallography at 2.95-Å resolution (Supplementary Table 3). Symmetry-related hexamers abut against each other in the crystal lattice, deforming the planar arrangement of the six subunits in each molecule (Supplementary Fig. 1j, k). This produces an "open-ring" conformation in which the subunits at the loose A/B interface are significantly splayed apart, and the small subdomain of the F subunit becomes highly disordered (Supplementary Fig. 1k, l). The individual TgMcrB^AAA monomers, however, adopt the same overall conformation and organization of nucleotide binding, as is observed in the cryo-EM reconstruction (Supplementary Fig. 1m–o). The distorted appearance of the crystallographic hexamer suggests a greater flexibility at the loose interfaces, which could more readily be influenced by crystal packing forces.

**TgMcrB^AAA contains the complete machinery for nucleotide hydrolysis.** Nucleotide hydrolases harness the energy of ATP or GTP hydrolysis to catalyze energetically unfavorable biological reactions, coordinate signal transduction events, and power protein conformational changes that orchestrate a multitude of cellular processes[45]. Efficient hydrolysis requires (i) the binding and recognition of the appropriate nucleotide substrate, (ii) the correct positioning of a water molecule for an in-line $S_N2$ attack on the γ-phosphate to initiate cleavage of the phosphoanhydride bond, and (iii) neutralization of a negative charge that develops between the β- and γ-phosphates in the transition state[46].

While a conserved sequence motif of GxxGxGK[T/S] (P-loop/ Walker A motif) coordinates the α- and β-phosphates in both ATPases and GTPases[47], the remaining catalytic machinery, specificity determinants, and charge-compensating elements vary from enzyme to enzyme. AAA+ proteins contain four additional sequence motifs—Walker B, Sensor I, Sensor II, and second region of homology (SRH)—that contribute to ATP binding and hydrolysis along with the conserved P-loop/Walker A motif[33,48]. The Walker B motif (D[D/E]xx) stabilizes an essential magnesium cofactor and acts in concert with a polar residue in the Sensor I motif to orient the catalytic water for nucleophilic attack on the γ-phosphate. The Sensor II motif localizes to helix 7 and contains a conserved arginine that interacts with the γ-phosphate. By convention, the subunit contributing these structural motifs to the nucleotide-binding pocket is referred to as the *cis* subunit. The neighboring, *trans* subunit inserts the arginine finger at the end of helix 4 of the SRH into the nucleotide-binding pocket, where it stabilizes the γ-phosphate and contributes to the charge compensation in the transition state.

Each composite active site of TgMcrB^AAA contains one GTPγS molecule and a bound magnesium ion (Fig. 2a–c). In the *cis* subunit, the main-chain atoms of the Walker A motif interact with the α- and β-phosphates, and Lys221 contacts the γ-phosphate of GTPγS (Fig. 2a, b). Thr222 (Walker A) and Asp356 (Walker B) coordinate the magnesium cofactor along with two ordered water molecules (Fig. 2a). Mutation of these conserved side chains to alanine impairs the basal GTPase activity of TgMcrB^AAA (Fig. 2c, d). Glu537 (Walker B) lies in close

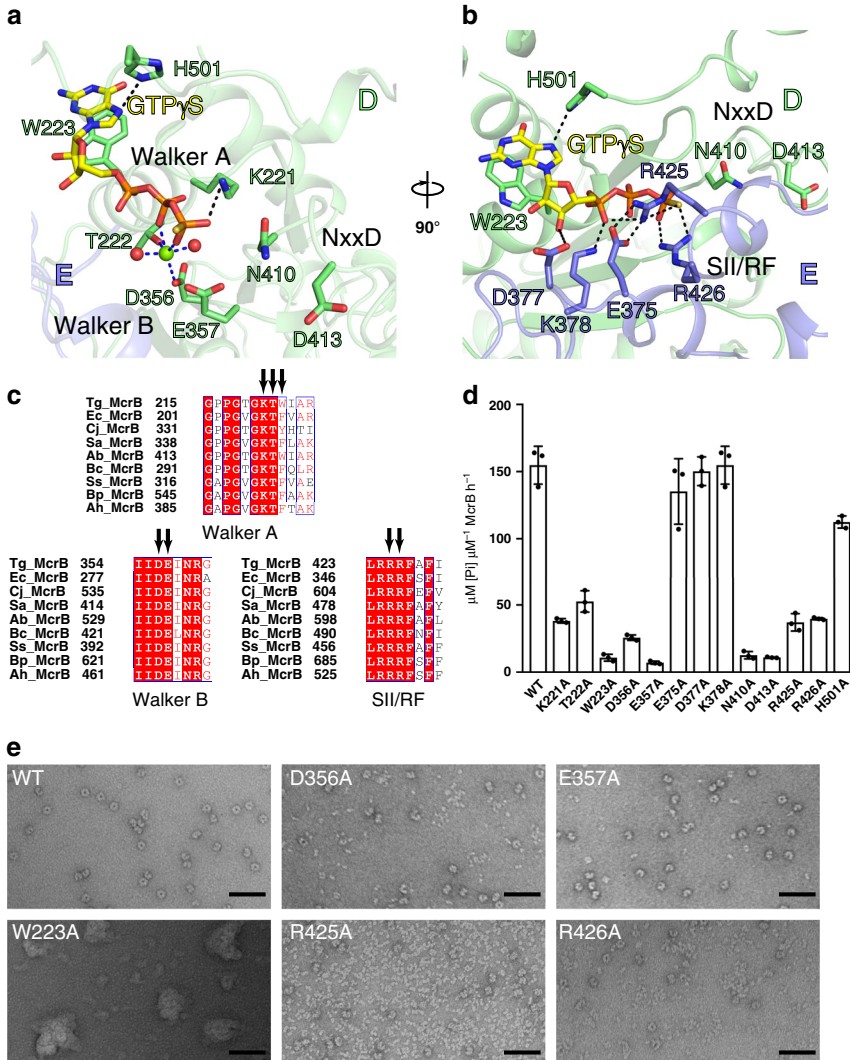

**Fig. 2 Catalytic residues involved in the basal GTPase activity of TgMcrB. a, b** Close-up views of the GTP-binding site at the tight D/E interface, highlighting residues involved in *cis* interactions, in particular those of the Walker A and B motifs, and the NxxD motif (**a**), and residues involved in *trans* interactions, in particular those of the Sensor II/arginine finger (SII/RF) motif (**b**). Spheres indicate waters (red) and a magnesium ion (green). Dashed lines indicate hydrogen bonds (black) and metal coordination (blue). **c** Sequence alignment of McrB homologs for the classic Walker A and B motifs, and the SII/RF motif. Arrows indicate the catalytic residues. Sequence alignment abbreviations are as follows: Tg *Thermococcus gammatolerans*, Ec *Escherichia coli,* Cj *Campylobacter jejuni*, Sa *Staphylococcus aureus*, Ab *Aciduliprofundum boonei,* Bc *Bacillus cereus,* Ss *Streptococcus suis*, Bp *Butyrivibrio proteoclasticus*, Ah *Anaerobutyricum hallii*. **d** Basal GTPase activity of wild-type TgMcrB and alanine mutants at the residues shown in **a** and **b** (n = 3, mean ± standard deviation). **e** Selected micrograph areas of negatively stained wild-type and mutant TgMcrB$^{AAA}$ incubated with GTPγS. Scale bars are 50 nm.

proximity to the γ-phosphate, primed to help stabilize a catalytic water (Fig. 2a). An alanine substitution at this position completely abolishes hydrolysis activity (Fig. 2d). Negative stain EM indicates that the Asp356Ala mutation has a higher propensity to disrupt the TgMcrB$^{AAA}$ hexamer than the Glu357Ala mutation (Fig. 2e), consistent with their distinct functions in nucleotide binding/stabilization versus catalysis. This result mirrors the oligomerization defects observed in EcMcrB when the corresponding residues (Asp279 and Glu280) were mutated[20], suggesting that the aspartate residue functions in magnesium binding, while the glutamate residue is critical for coordinating the catalytic water as in other AAA+ proteins[33]. Notably, the conserved McrB consensus loop ($^{409}$MNxxDR$^{414}$) replaces Sensor I and is located close to the γ-phosphate (Fig. 2 and Supplementary Fig. 2). Asn410Ala and Asp413Ala mutants significantly impair basal GTPase activity (Fig. 2d), suggesting they are critical for catalytic turnover, rather than for nucleotide binding, as was previously predicted[35].

His501 and Trp223 in the *cis* subunit sandwich the guanine base of GTPγS (Fig. 2a). His501 is situated above and forms a hydrogen bond with the 7′ nitrogen. Trp223, which lies adjacent to the Walker A motif, forms an unusual parallel π-stacking interaction from below that is absent in the majority of both GTPases and AAA+ proteins[44,49], but has recently been observed in the YCJK stress protein[50]. Mutation of Trp223 to Ala completely abolishes the basal GTPase activity (Fig. 2d) and causes the protein to aggregate, as seen by negative stain EM imaging (Fig. 2e). These observations indicate that π-stacking is critical for both McrB GTP binding and the stability of the oligomeric assembly. Although the aromatic residue is not strictly conserved across the McrB family, every homolog contains a residue at this position that is capable of π-stacking (Trp, Phe, Tyr, or Arg; Fig. 2c and Supplementary Fig. 2), including Phe209 in EcMcrB[41].

The *trans* subunit also contributes numerous conserved side chains that stabilize different portions of the bound nucleotide.

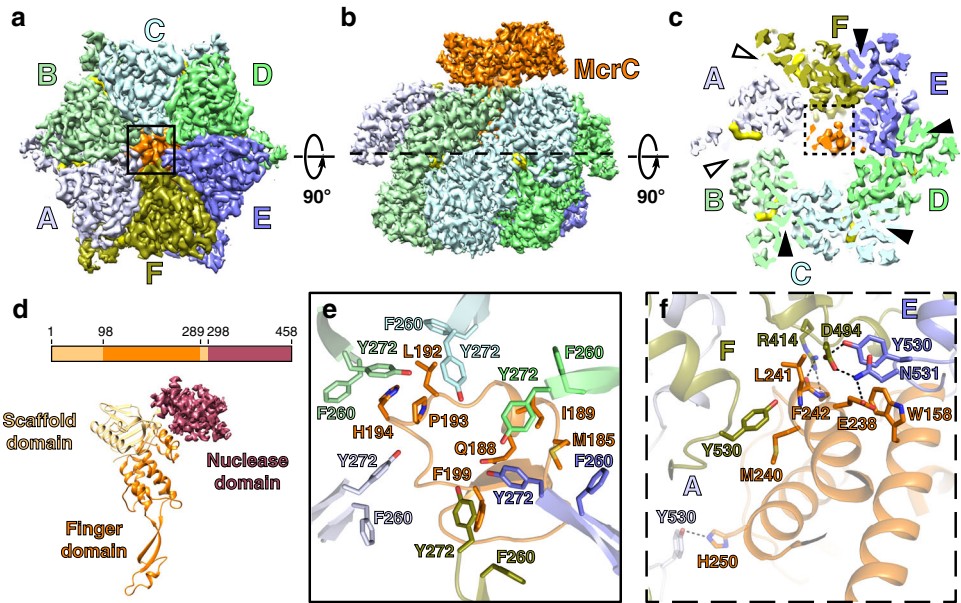

**Fig. 3 Asymmetric assembly of the TgMcrBC complex. a, b** Bottom and side views of the cryo-EM density map of the TgMcrBC half-complex. TgMcrB subunits are colored as in Fig. 1, TgMcrC is shown in orange, and nucleotides in yellow. **c** Slice section through the map of the TgMcrBC half-complex at the level of the bound nucleotides, indicated by the dashed line in **b**. Solid and empty arrowheads indicate tight and loose interfaces, respectively. **d** Domain architecture of TgMcrC. **e** Close-up view of the interactions of TgMcrC with TgMcrB at the bottom of the hexamer, indicated by the black square in **a**. **f** Close-up view of the interactions of TgMcrC with the TgMcrB hexamer at the E/F and F/A interfaces, indicated by the dashed black square in **c**. Dashed lines indicate hydrogen bonds.

Asp377 interacts with the 3′ ribose hydroxyl group, while Glu375 and Lys378 coordinate the α-phosphate (Fig. 2b and Supplementary Fig. 2). Mutations of these side chains had negligible effects on basal GTPase activity (Fig. 2d). Arg426 in helix α11 acts as the charge-compensating arginine finger, here forming hydrogen bonds with the γ-phosphate in the ground state (Fig. 2b). A second neighboring arginine located on the same helix, Arg425, assumes the role of the missing Sensor II motif (Fig. 2b, c). Arg425Ala and Arg426Ala mutations impair basal GTPase activity, and disrupt hexamer formation (Fig. 2d, e). All the *trans* interactions with GTPγS are prominent at the tight interfaces but are lost at the loose interfaces. Since the cryo-EM density for the Arg side chains in the Sensor II/arginine finger motif are also weaker at the loose interfaces (Supplementary Fig. 1o), the GTP-binding sites at these locations are likely in a non-catalytic state. Taken together, these results indicate that TgMcrB[AAA] possesses all the critical residues needed to bind and hydrolyze GTP.

**TgMcrB and TgMcrC form an asymmetric complex**. We next sought to elucidate structural and biochemical consequences of TgMcrC binding to TgMcrB. Purified TgMcrC was very sensitive to buffer conditions and could only be concentrated in the presence of TgMcrB. Together full-length TgMcrB and TgMcrC formed stable, dumbbell-shaped complexes in the presence of GTPγS that were suitable for structure determination by cryo-EM (Supplementary Fig. 4a). Initial image processing showed that the complex consists of two TgMcrB hexamers connected through TgMcrC dimerization (Supplementary Fig. 4b). Because of structural variability, however, we were only able to refine a "half"-complex (Supplementary Fig. 4c), which yielded a map at an overall resolution of 2.4 Å (Fig. 3 and Supplementary Fig. 4d–g). In this reconstruction, a single TgMcrC binds the TgMcrB hexamer by inserting itself through the central pore of the AAA+ ring in an asymmetric fashion (Fig. 3a–c).

The resolution of our reconstruction allowed us to build the TgMcrC structure de novo. Each monomer contains a scaffold domain, a "finger domain" and a C-terminal endonuclease domain (Fig. 3d and Supplementary Fig. 5). The scaffold domain (residues 1–98 and 289–298) consists of a barrel-like structure that centrally positions the two flanking domains, forming a rigid connection between the finger and endonuclease domains. The finger domain (residues 99–288) adopts an extended, segmented structure with two antiparallel helices that contact the nuclease domain above, a helical bundle, and a long β-sheet "stalk" that protrudes downward, terminating in a loop-helix-loop region at the tip (Fig. 3d and Supplementary Fig. 6a). The C-terminal endonuclease domain (residues 299–458) rests atop the structure and though poorly resolved in our map exhibits a fold characteristic of PD-(D/E)xK family enzymes.

The finger domain spans the entire length of the hexamer and its binding interface changes along the axis of the central pore (Supplementary Fig. 6a–e). At the top of the ring, the helical bundle associates with the F and E subunits, and then tilts to contact the E and D subunits near the middle of the assembly (Fig. 3c and Supplementary Fig. 6a–c). We also observe interactions between the β-sheet stalk and the E subunit at this midpoint (Supplementary Fig. 6a, d). The loop-helix-loop at the distal tip of the finger domain plugs a narrow opening at the very bottom of the McrB hexamer (Fig. 3a and Supplementary Figs. 5 and 6a, e). Conserved aromatic residues Phe260 and Tyr272 from the helix 2 inserts of each McrB subunit surround and stabilize the tip (Fig. 3e and Supplementary Fig. 2). While the finger domain interacts with all six subunits of TgMcrB at the bottom of the hexamer, TgMcrC binds the hexamer in a highly asymmetric fashion.

TgMcrC binding breaks the parallel π-stacking interaction between Arg414[F] and Tyr530[E] at the E/F interface (Fig. 3c, f), which has the smallest interaction area among the four tight interfaces (~2400 Å² versus >2500 Å² for all the others). This perturbation changes the conformation of the 414–420 loop in subunit F as Arg414[F] rotates to hydrogen bond with the main-chain atoms of Leu241[McrC] and Phe242[McrC] (Fig. 3f and

Supplementary Fig. 6h, i). Concomitantly, Tyr530 and Asn531 in subunit E hydrogen bond to Asp494 in subunit F. Glu238 in the finger domain further stabilizes this conformation through an additional hydrogen bond with Asp494$^F$. TgMcrC binding also generates some additional interactions in the F/A interface, where His250$^{McrC}$ hydrogen bonds with Tyr530$^{McrB}$ from the A subunit, and Met240$^{McrC}$ and Leu241$^{McrC}$ form van der Waals interactions with Tyr530$^{McrB}$ in the F subunit (Fig. 3f and Supplementary Fig. 5). These interactions, which bury a combined surface area of 1298 Å$^2$, serve to anchor McrC at the top of the ring, restricting its motion and orientation. Despite the localized differences at the E/F interface, the conformation of the TgMcrB hexamer remains largely unchanged in the TgMcrBC complex (overall RMSD of 0.75 Å compared to TgMcrB$^{AAA}$ alone, based on corresponding Cα atoms), with its intrinsic asymmetry, and the remaining tight and loose interface interactions preserved (Fig. 3c and Supplementary Fig. 6f, g). These findings indicate that TgMcrC does not induce substantial remodeling of the TgMcrB hexamer, but instead adapts and exploits its intrinsic asymmetry when binding.

**TgMcrC binding optimally positions existing catalytic machinery to stimulate GTP hydrolysis.** A distinguishing feature of the *E. coli* McrBC system is the ability of McrC to stimulate McrB's GTP hydrolysis in vitro[35,51]. Purified TgMcrC similarly stimulates TgMcrB's basal GTPase activity, demonstrating that this is also a conserved property of other homologs (Fig. 4d). Our high-resolution TgMcrBC structure reveals the underlying molecular mechanism governing this stimulation. As a consequence of the structural asymmetry imposed by the TgMcrB hexamer, TgMcrC's finger domain engages only a single active site at a time (Fig. 4a). Here, the helical bundle wedges against the D/E interface and inserts a highly conserved arginine (Arg263$^{McrC}$) at the edge of the pocket (Fig. 4b, c and Supplementary Fig. 6a, c). Acting through a hydrogen-bonding network that includes Asn359$^{McrB}$, Asn410$^{McrB}$, Asp413$^{McrB}$, and a bridging water (H$_2$O$^{Bridge}$), Arg263$^{McrC}$ ultimately alters the conformation of the McrB consensus loop (Supplementary Fig. 2). This reorganization allows Asn410 and E357 of the Walker B motif to position a second water (H$_2$O$^{Cat}$) that is poised for nucleophilic attack on the γ-phosphate (Fig. 4b). Glu357 also acts in concert with the Asp356 of the Walker B motif to stabilize a third water molecule that completes the octahedral coordination of the magnesium cofactor (Fig. 4b).

Alanine substitutions at Asn410 and Asp413 in full-length TgMcrB abolish both basal and McrC-stimulated GTPase activity (Fig. 4d), underscoring their crucial catalytic function. Mutation of Arg263$^{McrC}$ to alanine selectively abrogates the stimulatory effect of McrC binding without impairing basal turnover (Fig. 4d). The apparent GAP function thus arises from an indirect reconfiguration of the side chains that orient the catalytic water rather than promoting charge compensation in the transitions state.

**Sequential rearrangements of the consensus loop control the cycle of McrB GTP hydrolysis.** The consensus loop and charge-compensating arginine finger (Arg426$_{trans}$) adopt different conformations at each of the six interfaces within the McrC-bound TgMcrB hexamer (Fig. 4e–j). As described above, the tight D/E interface shows an McrC-activated conformation with Arg426$_{trans}$ stabilizing the γ-phosphate and Asn410 properly arranged to orient the catalytic water (Fig. 4b, e). In the adjacent tight E/F interface, Asn410 and Arg426$_{trans}$ appear in close contact with the γ-phosphate of GTPγS in a manner that excludes a potential catalytic water (Fig. 4f). The loose F/A interface uniquely contains

GDP with the side-chain oxygen of Asn410 forming a hydrogen bond with the β-phosphate. This partially occludes the space normally occupied by the γ-phosphate and forces Arg426$_{trans}$ into a conformation in which it is angled away from the nucleotide (Fig. 4g). At the loose A/B interface, Arg426$_{trans}$ and Asn410 both point away from GTPγS, likely a consequence of the weakened inter-subunit interactions (Fig. 4h). In both the tight B/C and C/D interfaces, Arg426$_{trans}$ interacts with the γ-phosphate, but Asn410 faces away from the nucleotide (Fig. 4i, j). These pockets appear primed for hydrolysis but unable to proceed efficiently, as Asp413 adopts random orientations in the absence of McrC, and thus cannot help stably redirect Asn410 to position the catalytic water (Fig. 4i, j).

These conformational differences likely reflect different states in the hydrolysis cycle, with the B/C and C/D active sites occupying a GTP-bound, pre-hydrolysis state, D/E most likely the activated transition state, E/F assuming a post-hydrolysis state, and the loose GDP-bound F/A and GTPγS-bound A/B sites, depicting the phosphate release and subsequent nucleotide exchange steps, respectively. Together these data imply that TgMcrB GTP hydrolysis proceeds through a coordinated, sequential mechanism.

**McrBC homologs share a conserved architecture and catalytic mechanism.** To establish whether different homologs use a conserved mechanism for stimulated hydrolysis, we determined the single-particle cryo-EM structure of the complex formed by the full-length *E. coli* proteins (EcMcrBC) in the presence of GTPγS. EcMcrBC also formed dumbbell-shaped particles and we refined a "half"-map reconstruction of these assemblies to an overall resolution of 3.3 Å (Fig. 5 and Supplementary Fig. 7). The half-complex structure shares the same overall asymmetric architecture as the previously reported structure of the truncated *E. coli* restriction complex that lacks the N-terminal domain of McrB (EcMcrBΔNC)[41] (Supplementary Fig. 8). Despite being stabilized by different guanine nucleotide analogs (5′-guanylyl imidodiphosphate (GMPPNP) versus GTPγS), the two models superimpose with an overall RMSD of 2.97 Å, even across the asymmetrically interacting McrC subunit (Supplementary Fig. 8a). The orientations of interacting subunits are also spatially conserved (Supplementary Fig. 8c–e), suggesting that the assembly and asymmetric architecture of the restriction complex are fundamentally maintained, regardless of the used nucleotide analog.

As with TgMcrBC, a single EcMcrC monomer inserts into the central pore of the EcMcrB hexamer (Fig. 5a, b). A cross-section slice through the map at the height of the bound nucleotides reveals that the same intrinsic asymmetry is present, with loose F/A and A/B interfaces, and tight B/C, C/D, D/E and E/F interfaces (Fig. 5c). The unique interactions stabilizing each tight interface are also conserved in EcMcrBC and absent in the loose interfaces: Arg337 (Arg414 in TgMcrB) and Asp343 (Asp420 in TgMcrB) interact in *trans* with Phe428 (Tyr530 in TgMcrB) and Arg283 (Arg360 in TgMcrB), respectively (Supplementary Fig. 9a, b).

EcMcrC shares the same general architecture as TgMcrC, featuring an extended finger domain and a C-terminal nuclease domain (Supplementary Fig. 9c). EcMcrC, however, lacks the N-terminal portion of the scaffold domain, retaining only a small β-hairpin insertion between the finger and nuclease domains (Fig. 3d and Supplementary Fig. 9c). Sequence alignment of McrC family proteins suggests that these insertion strands serve as a conserved linker between the finger and nuclease domains (Supplementary Fig. 5). The finger domains superimpose with an RMSD of 2.4 Å (sequence identity: 20%), confirming the

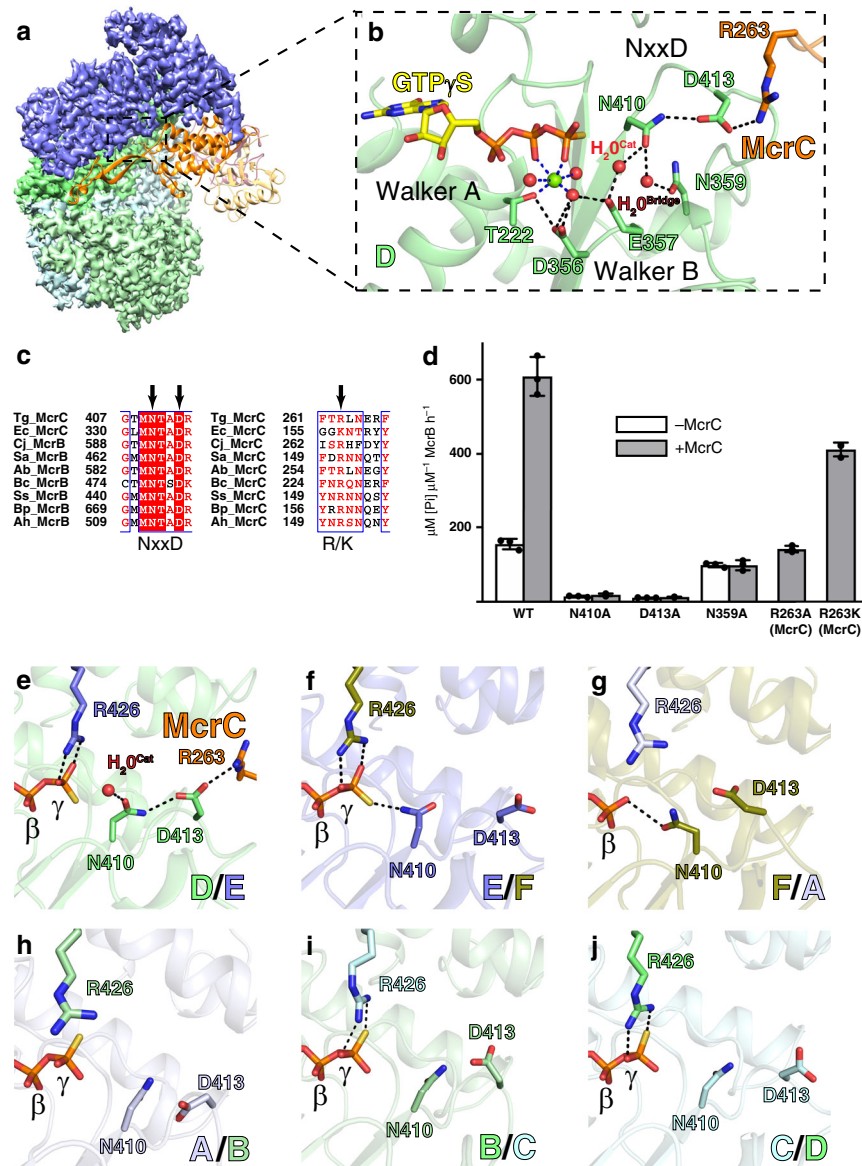

**Fig. 4 Structural basis for TgMcrC-mediated stimulation of TgMcrB GTPase activity. a** Side view showing the interaction of TgMcrC with the D/E interface of the TgMcrB hexamer. TgMcrB and TgMcrC are colored as in Fig. 3, and shown in surface and ribbon representation, respectively. For clarity, subunits A and F are not shown. **b** Hydrogen-bonding network formed by TgMcrC with residues of the NxxD motif at the D/E interface of the TgMcrB hexamer. Spheres indicate waters (red) and a magnesium ion (green). Dashed lines indicate hydrogen bonds (black) and metal coordination (blue). For clarity, the *trans* interacting residues in subunit E are not shown. **c** Sequence alignment of McrB and McrC homologs for the McrB signature sequence (NxxD), and the region in McrC that contains the inserted arginine/lysine residue (R/K). Abbreviations for the aligned species are as in Fig. 2c. **d** Basal (−McrC) and TgMcrC-stimulated (+McrC) GTPase activity of TgMcrB for wild-type proteins and mutants, with single amino acid substitutions either of residues around the NxxD motif in TgMcrB or of residues in TgMcrC (n = 3, mean ± standard deviation). **e–j** Arrangement of the asparagine and aspartate residues of the NxxD motif at the six interfaces in the TgMcrB hexamer of the TgMcrBC complex.

overall structural conservation between these evolutionarily remote homologs.

The structural asymmetry present in the EcMcrBC complex similarly biases EcMcrC to associate with only a single active site at a time (Fig. 5c, d). EcMcrC inserts Lys157 into the D/E interface of the EcMcrB hexamer, and employs the same hydrogen-bonding network seen in the TgMcrBC complex to reorient Asn333 and Asp336 in the McrB signature motif (Fig. 5e). Although we do not resolve the catalytic or bridging waters in our structure of the *E. coli* complex, the cryo-EM density supports the location of the Lys157 side chain (Supplementary Fig. 9d–f). Lys157 was modeled further away from the signature motif in the EcMcrBΔNC structure[41], possibly

owing to weaker density and the lower resolution of the map. The Cα positions of this residue and other critical active-site components align with those in our reconstruction (Supplementary Fig. 8b). Asn282 spatially occupies the same position as Asn359 in TgMcrB (Figs. 4b and 5e). The rest of the catalytic machinery is also conserved (Fig. 5e and Supplementary Fig. 9g).

Our structural findings rationalize previous phenotypes associated with consensus loop mutants in EcMcrB. Asn333Ala and Asp336Asn substitutions would disrupt the hydrogen-bonding network needed to position the catalytic water, leading to a complete loss of GTPase activity and the abrogation of DNA cleavage, when translocation is required to engage complexes bound at distant R$^{MC}$ sites[16,35]. Loss of stimulated GTPase

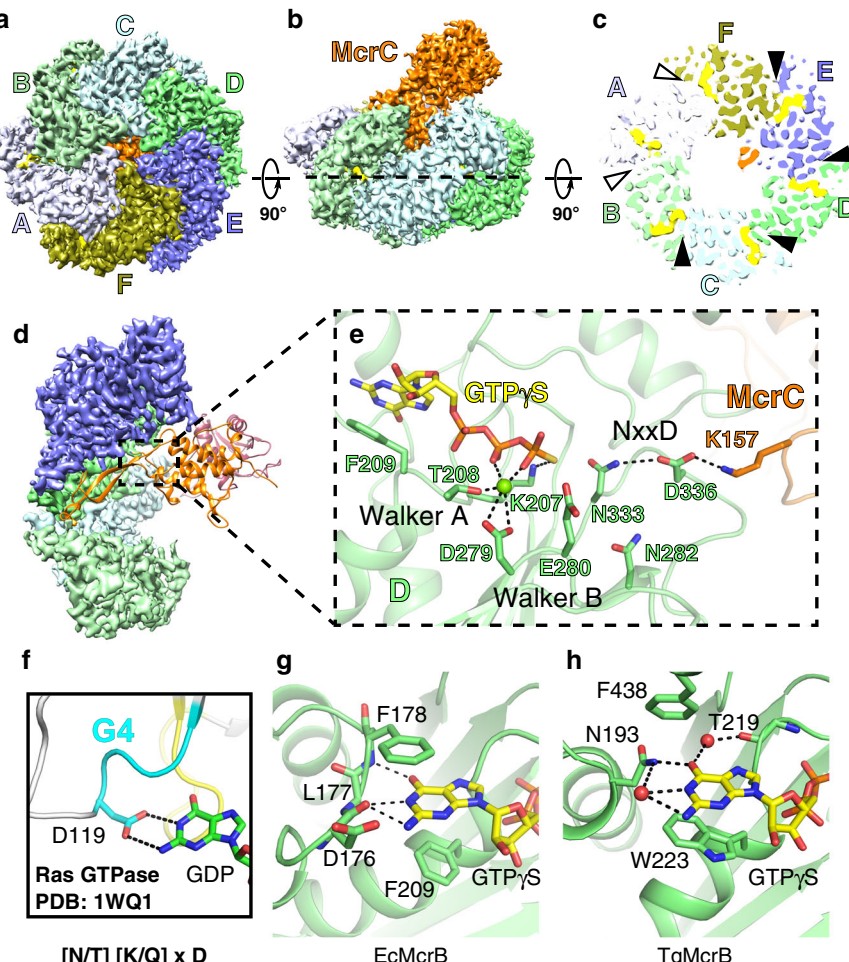

**Fig. 5 The McrBC complexes of *E. coli* and *T. gammatolerans* show a conserved architecture. a**, **b** Bottom and side views of the cryo-EM density map of the EcMcrBC half-complex. **c** Slice section through the map of the EcMcrBC half-complex at the level of the bound nucleotides, indicated by the dashed line in **b**. Solid and empty arrowheads indicate tight and loose interfaces, respectively. **d** Close-up view of the interaction of EcMcrC with the EcMcrB hexamer at the D/E interface. EcMcrB and EcMcrC are shown in surface and ribbon representation, respectively. For clarity, subunits A and F are not shown. **e** Hydrogen-bonding network formed by EcMcrC with residues of the NxxD motif at the D/E interface of the EcMcrB hexamer. Spheres indicate waters (red) and a magnesium ion (green). Dashed lines indicate hydrogen bonds and metal coordination. **f–h** Structural basis for guanine recognition in the Ras GTPase (PDB: 1WQ1)[68] and in the McrB homologs (EcMcrB and TgMcrB).

activity due to an alanine mutation at Asn282 would arise from a similar structural perturbation[16]. Interestingly, substituting a lysine for TgMcrC's catalytic Arg263 partially restores the stimulatory effect that is lost when this side chain is replaced with alanine (Fig. 4d). Together these data demonstrate that stimulated GTP hydrolysis in different McrBC homologs occurs via a conserved molecular mechanism.

**Divergent McrB homologs employ the same generalized principles for nucleotide specificity.** In every GTPase, the conserved sequence [N/T] [K/Q]xD (termed the "G4 element") confers nucleotide specificity[37,40,49]. The absolutely conserved aspartate side chain in this motif forms specific hydrogen bonds with the 1′ amine and 2′ amino group of the guanine base, thereby distinguishing it from ATP (Fig. 5f). Nothing in the TgMcrB AAA+ domain makes contact with this portion of the nucleotide (Fig. 2a, b), suggesting other structural features fulfill this role. Our reconstructions of the full-length EcMcrBC and TgMcrBC complexes reveal how each individually achieves this end (Fig. 5g, h). In EcMcrBC, a loop that lies directly upstream of the AAA+ domain coordinates the guanine base through main-chain interactions (Fig. 5g). The backbone carbonyl of Asp176 hydrogen

bonds with both the 1′ amine and 2′ amino group of the guanine base, while the main-chain nitrogen of Phe178 reads out the 6′ carbonyl group. The same hydrogen bonds were observed in the truncated, GMPPNP-stabilized EcMcrBΔNC complex containing residues 162–465[41]. Collectively these interactions would discriminate against the substitution of an amino group at the 6′ position (as in ATP and XTP) and the absence of an amino group at the 2′ position (as in ATP and ITP), consistent with EcMcrB's nucleotide selectivity preferences of GTP > ITP > XTP >> ATP[51]. TgMcrB, in contrast, specifically coordinates the guanine base through two water-mediated interactions (Fig. 5h). Asn193 at the very beginning of the AAA+ domain directly hydrogen bonds to guanine's 6′ carbonyl and orients a water molecule to interact with the 1′ amine and 2′ amino group. The backbone carbonyl of Thr219 also interacts with the 6′ carbonyl group of the base via a second bridging water. Importantly, the fundamental chemistry underlying guanine nucleotide recognition is conserved between both homologs despite each utilizing different structural elements.

**McrBC forms a tetradecameric assembly through the dimerization of McrC.** Previous studies reported that EcMcrBC

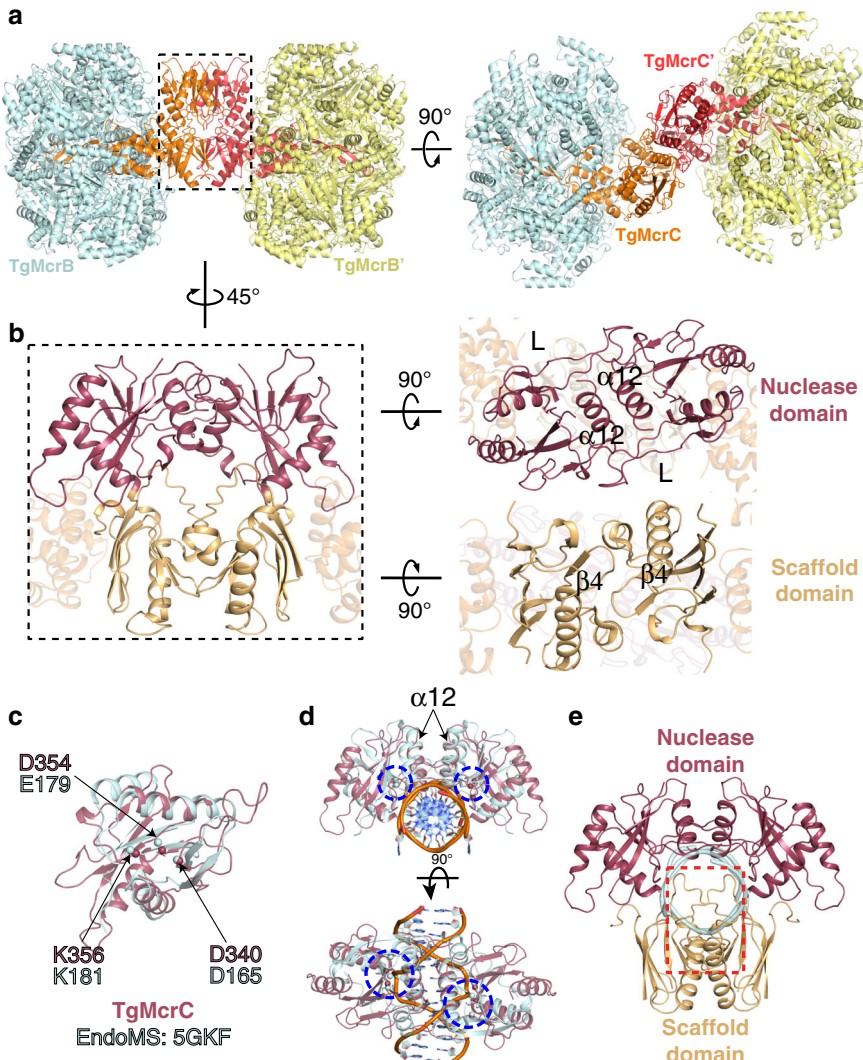

**Fig. 6 The tetradecameric assembly of the TgMcrB^AAAC complex shows a cleavage-incompetent conformation. a** Side views of the full TgMcrB^AAAC complex are shown in ribbon representation. Two TgMcrC (orange and red) form a dimer that bridges two TgMcrB hexamers (cyan and yellow). **b** Close-up views of the TgMcrC dimer interfaces formed by the two nuclease domains (upper right panel) and the two N-terminal domains (lower right panel). The α12 helix and a loop between the β10 and β11 strands are labeled as "α12" and "L", respectively. **c** Superposition of the monomeric structures of TgMcrC and EndoMS (sequence identity: 12%; PDB: 5GKF)[53]. The conserved residues involved in the cleavage activity are labeled and shown as spheres. **d** Structural comparison between the TgMcrC dimer in the TgMcrB^AAAC complex and the EndoMS dimer in a DNA-bound state. The blue dashed circles indicate the active sites for DNA cleavage. **e** Illustration of the cleavage-incompetent conformation of TgMcrC. For clarity, the structure of the EndoMS protein is not shown. The backbone of the DNA substrate bound to EndoMS is colored in cyan. The red square indicates the regions of potential steric clashes.

complexes form tetradecameric assemblies in vitro[20,21,41]. In our hands, dimeric McrBC complexes generated using the full-length Tg and Ec proteins exhibit a high degree of conformational variability, which hampered efforts to calculate complete, interpretable maps for these larger oligomeric states. To overcome this limitation, we produced complexes containing full-length TgMcrC bound to the AAA+ domain of TgMcrB (TgMcrB^AAAC) in the presence of GTPγS. This assembly was structurally more homogeneous and allowed us to calculate maps of the "half"-complex at 3.7-Å resolution, as well as a C2-symmetrized map of the entire TgMcrB^AAAC tetradecameric complex at 4.2-Å resolution (Supplementary Fig. 10). A TgMcrC dimer bridges two TgMcrB^AAA hexamers in this structure (Fig. 6a), with the scaffold and nuclease domains forming the dimer interface (Fig. 6b). The nuclease domains associate through their α12 helices and a loop between the β10 and β11 strands ("L"), whereas the neighboring scaffold domains interact with each other through their β4 strands that form main-chain

hydrogen bonds with each other. EcMcrBΔNC shows a similar overall arrangement, although numerous single-particle classes with different angles between the two half-complexes were reported for this assembly[41]. Interestingly, the half-complex reconstruction of the full-length EcMcrBC tetradecamer shows density for an additional ordered EcMcrC nuclease domain (Supplementary Fig. 11a). The organization of the EcMcrC nuclease domains at this dimer interface is identical to that seen in other McrBC complexes[41] (Fig. 6a, b), with the α10 helix and an analogous extended loop serving as the primary points of contact (Supplementary Fig. 11b). This observation implies the same tetradecameric assembly is formed by the full-length construct.

**The McrC dimer adopts a cleavage-incompetent conformation in the absence of a DNA substrate.** The DNA-bound structures of other PD-(D/E)xK nucleases provide a template for modeling

McrC's cleavage activity. Of the many structural homologs identified by the DALI server[52], the coordinates of the *Thermococcus kodakarensis* EndoMS endonuclease[53] (sequence identity 12%; PDB: 5GKF; $Z$-score 7.9) provided the best framework for these purposes. EndoMS binds DNA as a dimer, with each active site attacking a single strand of the DNA duplex to induce a double-strand break. As with other PD-(D/E)xK enzymes, Asp165$^{EndoMS}$, Glu179$^{EndoMS}$, and Lys181$^{EndoMS}$ coordinate a divalent metal cofactor that is required for catalytic function[53,54]. Structural superposition confirms TgMcrC's C terminus shares the same fold and identifies Asp340$^{TgMcrC}$, Asp354$^{TgMcrC}$, and Lys356$^{TgMcrC}$ as putative catalytic side chains based on their spatial alignment with the EndoMS metal-binding residues (Fig. 6c). EcMcrC also shares this structural homology (Supplementary Fig. 11c). Importantly, our modeling is consistent with previous biochemical data showing that mutation of the predicted catalytic residues in EcMcrC (Asp224$^{EcMcrC}$, Asp257$^{EcMcrC}$, and Lys259$^{EcMcC}$) impairs cleavage of modified DNA in vitro[55]. Further comparison shows that the organization and location of the active sites in the TgMcrC and EcMcrC dimers is conserved between the two species (Supplementary Fig. 11d).

To gain insight into McrC cleavage, we overlaid two copies of the TgMcrC and EcMcrC endonuclease domains independently onto the dimeric, DNA-bound EndoMS complex (Fig. 6d and Supplementary Fig. 11e). The nuclease domains align in an orientation that resembles the dimer configuration captured in our cryo-EM structures; however, we observe numerous steric clashes in both models. TgMcrC's scaffold domain and the α12 nuclease helices collide with the DNA substrate (Fig. 6e). EcMcrC lacks an N-terminal scaffold domain yet still clashes with the DNA backbone, owing to the first helix of its nuclease domain being significantly longer (Supplementary Fig. 11f). Attempts to model similar interactions with other structurally related homologs like EcoRV[56] (PDB: 1AZ0; sequence identity: 8% with TgMcrC and 13% with EndoMS) and the *Sulfolobus solfataricus* Holliday junction endonuclease[57] (PDB: 1OB8; sequence identity: 17% with TgMcrC and 9% with EndoMS) resulted in substantial clashes between the two McrB hexamers. We therefore speculate that our dimeric McrBC structures depict a conformation that is incompatible with DNA cleavage, and that a major conformational change would be required for nuclease activity to proceed unencumbered.

## Discussion

Our structural analysis reveals that TgMcrB$^{AAA}$ forms an asymmetric hexamer, similar to the architecture adopted by many other AAA+ family proteins[58–65]. In the hexameric arrangement, four of the subunits, B, C, D, and E, occupy a radial sector (measured as the radius between the Cα positions of Lys221 residues in neighboring subunits) of 59°, with the other two subunits, A and F, occupying radial sectors of 60° and 64°, respectively. This distortion of the hexameric assembly, which results in four tight and two loose interfaces (Fig. 1c–e), appears to be maintained by the conformation of key interface residues—Arg360, Glu527, and Tyr530 in one monomer and Arg414, Asp420, and Arg424 in its neighbor—acting in *trans*. Alanine substitutions of these residues increase basal GTPase activity by ~two-fold (Supplementary Fig. 1i). We speculate that interface mutations alter the programmed asymmetry, causing the unrestrained individual subunits to wobble randomly and leading to uncoordinated, stochastic GTP hydrolysis throughout the hexamer. The asymmetry in the ring also explains how crystal packing forces could induce and/or sustain the open conformation observed in our TgMcrB$^{AAA}$ X-ray structure (Supplementary Fig. 1j–l), as the loose interfaces likely have a greater propensity

for flexibility resulting from the fewer stabilizing interactions. These observations argue that asymmetry is an intrinsic characteristic of the McrB$^{AAA}$ hexamer rather than being induced upon McrC binding, as has recently been proposed[41].

While all McrBC structures presented here display the same arrangement of four tight interfaces and two loose interfaces, the previous EcMcrBΔNC structure showed three GMPPNP-bound interfaces and three GDP-bound interfaces in the McrB hexamer, which were likely to be tight and loose interfaces, respectively[41]. This discrepancy might be due to subtly different binding affinities for nucleotide analogs and the sensitivity of the EcMcrB assembly to nucleotide depletion. TgMcrB, in contrast, exists as stable hexamers even in the absence of any nucleotide (Supplementary Fig. 1a), suggesting that the balance between nucleotide affinity, occupancy, and structural integrity could affect the dynamics of McrB AAA rings.

The TgMcrB$^{AAA+}$ domain possesses all the catalytic machinery needed for nucleotide hydrolysis. We find that the canonical *cis*-acting Sensor II arginine is replaced with a *trans*-acting arginine (Arg425) that is positioned adjacent to the charge-compensating arginine finger (Arg426) in helix α11 (Fig. 2b). Our cryo-EM and X-ray structures of TgMcrB$^{AAA}$ reveal that Arg425 is not only important for stabilizing Glu375 in *cis* as predicted from the previous structures of *E. coli* complexes[41], but also interacts with the phosphates of GTP in *trans* (Fig. 2b). Asn410 (consensus loop) and Glu357 (Walker B motif) together position the catalytic water. We also note that Trp223 forms a crucial π-stacking interaction with the guanine base that is present in the EcMcrBC reconstructions[41] (Fig. 5e) and functionally conserved at the sequence level in other homologs. Perturbing any of these side chains reduces basal GTP hydrolysis of TgMcrB$^{AAA}$. Similar phenotypes were observed with the corresponding mutations in the *E. coli* protein[16], indicating that the basic catalytic machinery is hardwired into the McrB AAA+ fold across evolution.

We demonstrate that McrC-stimulated GTP hydrolysis is a broadly conserved property of the McrBC family and not simply a unique feature of the *E. coli* homolog (Fig. 4d)[16,35]. While this type of stimulation is uncommon among AAA+ proteins, it resembles the activation of small G proteins by their cognate GAPs. GAPs enhance catalytic turnover either by contributing essential catalytic residues in *trans* or by conformationally stabilizing and/or reorienting active site elements into an optimal configuration[66]. In nearly every case, these interactions affect the charge-compensating element[67]. RasGAP, for example, provides the arginine finger needed for Ras turnover, while RGS4 binding to G$_{iα1}$ reorients an existing arginine in the switch I motif[68,69]. A notable exception is RapGAP, which provides in *trans* an asparagine that positions the nucleophilic water[70]. Our structures show that TgMcrC and EcMcrC stimulate hydrolysis indirectly by altering the conformation of the McrB consensus loop. Both proteins insert a conserved basic residue (Arg263$^{TgMcrC}$ and Lys157$^{EcMCrC}$) at the edge of the McrB active site and, via a hydrogen-bonding network, reposition a conserved asparagine (Asn410$^{TgMcrB}$ and Asn333$^{EcMcrB}$) that in turn correctly orients the catalytic water for nucleophilic attack on the γ-phosphate (Figs. 4b and 5e, and Supplementary Fig. 9e, f). This conserved molecular mechanism thus represents a unique variation on a common theme. We note that the helical bundle of the McrC finger domain wedges into the E/F interface at the top of the McrB hexamer in both structures (Fig. 3f). This interaction not only anchors McrC, but also directs its catalytic machinery to the adjacent active site at the D/E interface (Fig. 4b and Supplementary Fig. 6c). These constraints dictate that McrC stimulation can only occur at a single active site at any given time.

In our structures, the consensus loop and the in *trans* arginine finger adopt different conformations in each active site around

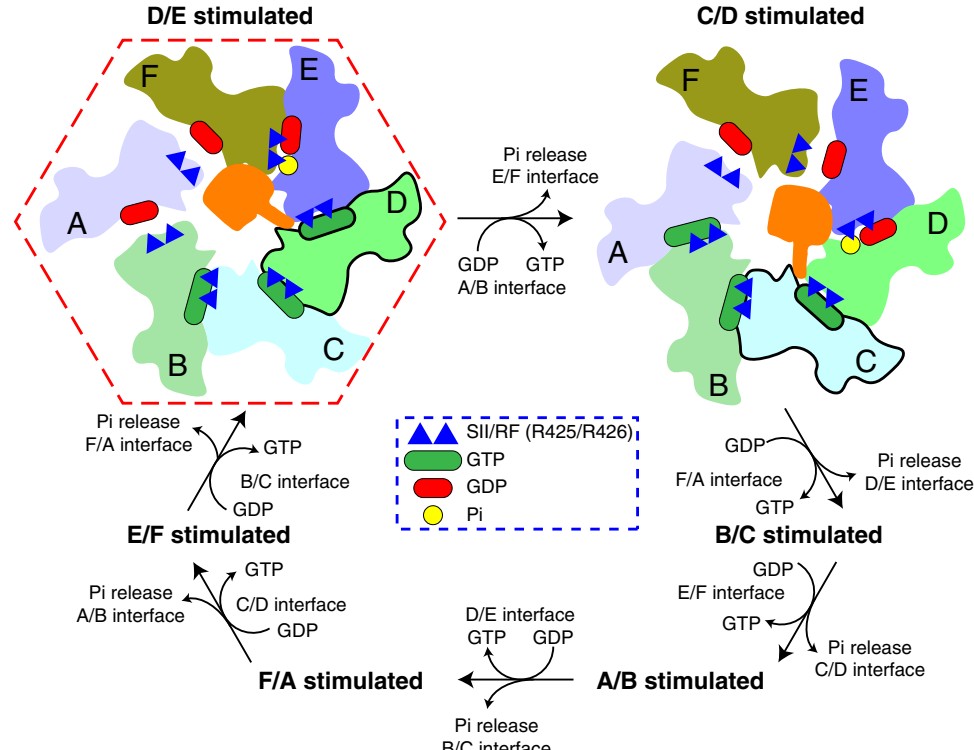

**Fig. 7 Rotation model for the catalytic cycle of McrB.** Schematic representation of the putative GTP hydrolysis cycle that proceeds sequentially in a clockwise manner around the hexameric McrB ring relative to McrC in the central pore. The "finger" extending from McrC represents the arginine/lysine residue that interacts with the NxxD motif. Ovals at the interfaces of the hexamer represent GTP (green) and GDP (red). The subunits indicated by the thick outlines are those in which the NxxD has been reorganized by McrC inserting its arginine/lysine residue. The red hexagon indicates the state observed in our cryo-EM structures.

the McrB hexamer (Fig. 4e–j). We interpret each configuration as representing a different state in the hydrolysis cycle (Fig. 7, dashed red outline). The McrC-engaged D/E active site assumes a transition state-like conformation with the catalytic machinery optimally positioned for stimulated turnover. In the tight C/D and B/C active sites, GTP is bound but the catalytic components are in a suboptimal conformation. This configuration suggests a pre-hydrolysis state that is primed for interaction with McrC. The loose A/B and F/A sites represent low-affinity, post-hydrolysis states that allow for free exchange of GTP and GDP, consistent with McrB not requiring a guanine nucleotide exchange factor. In support of this notion, we find GDP at the A/B site in our EcMcrBC structure (Supplementary Fig. 12a–f, s), the F/A site in the structure of the TgMcrBC complex (Supplementary Fig. 12g–l, t), and at both sites in the structure of the TgMcrB$^{AAA}$C complex (Supplementary Fig. 12m–r, u). The final tight E/F site likely adopts a post-hydrolysis state that is partially destabilized, but still remains intact due to the presence of the γ-phosphate in the bound GTPγS. These data suggest that McrC-stimulated GTP hydrolysis proceeds via a coordinated mechanism that cycles around the McrB hexamer, engaging each composite active site sequentially (Fig. 7). In this scheme, the release of the γ-phosphate and the intrinsic asymmetry of the complex serve as the driving forces for a rotational movement. Release of the phosphate would destabilize the E/F interface, converting it from a tight to a loose configuration. This could promote a transition of the A/B interface from loose to tight, where exchange of GTP for GDP has presumably occurred. Weakening the E/F interface would destabilize the interactions with the helical bundle that anchor the finger domain (Fig. 3f), thereby releasing McrC and allowing it to rotate. The asymmetry of the structure would bias the movement in a clockwise direction, as

the helical bundle of the finger domain would not be able to associate with the loose F/A interface, and thus would have to intercalate into the D/E interface. This engagement would orient McrC to insert its catalytic arginine/lysine into the C/D active site, where it could trigger the next hydrolysis event to power the motor (Fig. 7). The extensive contacts formed between the helix-loop-helix tip of the finger domain and all six subunits of the McrB hexamer (Fig. 3e) would ensure that McrC does not dissociate from the complex following stimulated turnover. The stepwise transition from one binding interface to the next (Fig. 7) is reminiscent of F/V-type ATPases[71–74].

A similar sequential mechanism for GTP hydrolysis and clockwise movement of McrC were previously proposed based on the EcMcrBΔNC structure[41]. In the half-complex of the EcMcrBΔNC tetradecameric structure, three GMPPNP and three GDP were assigned in the subunit interfaces of McrBΔN (Supplementary Fig. 12v). One of the GMPPNP-bound interfaces (the "CD interface", which corresponds to the E/F interface in this study) was assumed to be the McrC-stimulated active site. We interpret this interface as a post-hydrolysis site, and instead believe the stimulation and formation of the catalytic transition state occurs at the adjacent D/E interface. It was further speculated that the interaction of the "CD interface" with the β-sheet "stalk" of McrC-initiated GTP hydrolysis[41]. The resulting conformational changes in the McrB signature motif were not fully appreciated, however, due to the limited resolution of the EcMcrBΔNC structure. Our cryo-EM reconstruction of TgMcrBC unambiguously reveals the catalytic water molecules and illustrates how McrC's insertion of a basic residue specifically repositions the signature motif to trigger hydrolysis, providing a chemically and energetically favorable description of stimulated turnover. Given the conserved structural features and asymmetry

present in both the Tg and Ec complexes, we anticipate that other McrBC homologs will follow this mechanochemical model.

Efficient hydrolysis also depends on the ability of an enzyme to bind and differentiate its appropriate nucleotide substrate. GTPases use the conserved aspartate in the G4 element to coordinate substituents at the $1'$ and $2'$ positions of the guanine base, while AAA+ proteins recognize the amino group at the $6'$ position in adenine[37,68]. By reading out the $1'$, $2'$, and $6'$ positions of the guanine base, McrB homologs appear to have combined both strategies to fine-tune their specificity for GTP in the context of a AAA+ fold. Ec and TgMcrB both use the same basic chemistry for this recognition, but each employs different structural components to mediate these contacts (Fig. 5g, h). Interestingly, these pieces lie outside the core AAA+ fold and localize to either the flexible linker that connects to EcMcrB's N-terminal DNA-binding domain or the very start of helix α1 in TgMcrB (Supplementary Figs. 2 and 3a, colored in gold). Although the motor and cleavage machineries are conserved among McrB homologs (Figs. 3–5), the N-terminal domains and connecting linkers are highly divergent. Crystallographic studies have shown that EcMcrB uses the DUF3578 fold to bind methyl-cytosine modifications[17,19], whereas the N-terminal domain of TgMcrB consists of a YTH fold that specifically targets 6mA-modified DNA[29]. The related LlaJI restriction system from *Helicobacter pylori* binds DNA site-specifically via an N-terminal B3 domain[28]. The subtle distinctions we observe with regard to nucleotide recognition are therefore significant, and provide a blueprint for how divergent homologs can maintain the necessary pattern of hydrogen bonding even in radically different structural contexts. Future structural characterization will determine if these principles hold true for other McrBC family members.

Previous biochemical studies suggest that McrBC's stimulated GTP hydrolysis powers DNA translocation[18,22]. While we do not directly address how this may occur in this study, our structures impose constraints with regard to the potential pathway of DNA and the organization of a cleavage-competent McrBC complex. DNA and RNA typically pass through the central pore of hexameric AAA+ helicases and translocases driven by ATP hydrolysis[59,75]. Based on recent cryo-EM reconstructions, a similar mechanism has been proposed for EcMrBC, in which the McrB N-terminal domains might interact with DNA on the bottom of the hexamer and thread it into the central channel[41]. Although we see weak density in our full-length EcMcrBC map that corresponds to the N-terminal domains near the top of the complex (Supplementary Fig. 11g), numerous structural observations oppose this potential trajectory. First, McrC specifically binds in the center of the McrB hexamer, blocking access to this pathway in both the Tg and Ec complexes. The asymmetric association of the finger domain's helical bundle with the D/E/F subunits shrinks the pore diameter at the top of the hexamer to ~10 Å (Supplementary Fig. 6b), while the loop-helix-loop region completely occludes the pore at the bottom of the hexamer (Fig. 3a, and Supplementary Figs. 6e and 13a, b), which narrows to a diameter of ~8 Å even without McrC. Passage through the ring in this state would require both distortion and/or melting of the DNA duplex to conform to the narrow dimensions of the structure, as well as either a complete displacement or gross conformational reorganization of McrC. Such changes would uncouple the sequential, coordinated stimulation of GTP hydrolysis suggested by our structures and yield a translocation mechanism that would use a completely stochastic catalytic process, and would depend on alternating cycles of binding and dissociation for both McrC and DNA. While we cannot rule out that additional conformational changes occur upon DNA binding, biochemical characterization of EcMcrBC has shown that DNA binding and GTP hydrolysis are separate and distinct

properties in vitro[14,22,35,36]. It therefore seems unlikely that DNA binding would significantly alter the architectural and catalytic interactions that have been conserved across kingdoms. Second, we resolve clear density decorating the outside edges of the TgMcrB^AAA hexamer that we attribute to the TgMcrB N-terminal domains (Supplementary Fig. 11h). The localization of these domains nearly perpendicular to the pore axis would require DNA, if it were to pass through the center of the TgMcrBC complex, to bend dramatically, more than has been observed in any structure to date. Energetically, such a configuration would be extremely unfavorable[76]. The short seven amino acid linker connecting the N-terminal domains to the Tg AAA+ domains combined with the structural requirements of nucleotide selectivity would likely prohibit a large-scale rearrangement of these domains within the restriction complex. Taken together, these findings argue against a mechanism in which DNA passes through the central channel in the McrB hexamer. We speculate that McrBC complexes use an alternative, yet to be elucidated means to translocate DNA.

Both EcMcrBC and TgMcrBC form tetradecameric complexes that are bridged by an McrC dimer (Fig. 6a and Supplementary Fig. 11a). Our structural modeling, however, suggests that the conformation of this McrC dimer is incompatible with DNA binding and cleavage. Superposition with EndoMS shows that the N-terminal scaffold domain of TgMcrC and the first helix in the nuclease domain of EcMcrC clash with the modeled DNA substrate (Fig. 6d, e and Supplementary Fig. 11e, f). Modeling these interactions in the context of the tetradecameric complexes indicates further steric hindrance: superimposed DNA strands would clash with the McrB subunits (Supplementary Fig. 13c, d; left panels) and have a trajectory that is directed away from the central pore in the hexamer, offset by nearly 30° and 10° for the TgMcrB^AAA C and EcmMCrBΔNC structures, respectively (Supplementary Fig. 13c, d; right panels). These observations support the idea that the current structures represent binding/cleavage-incompatible conformations. It remains to be seen whether DNA binding alone could induce a cleavage-competent conformation. Interestingly, GTPγS does not support EcMcrBC DNA cleavage in vitro[22], consistent with our structural findings here. Moreover, mutation of Pro203 to valine in EcMcrB, a residue in a loop close to the γ-phosphate and the hexamer interface, significantly reduces both EcMcrC-stimulated GTP hydrolysis and DNA cleavage of an "ideal" substrate with R^MC sites optimally spaced 63 base pairs apart so as not to require translocation[22]. This finding raises the possibility that GTP hydrolysis is also needed for the transient reorganization of the McrC monomers, and that blocking this activity would lead to a nonproductive arrangement. Further experiments will be needed to fully understand how the McrBC complex cleaves DNA.

Modification-dependent restriction systems function as a conserved barrier to lytic phage infections. In the ongoing arms race between virus and host, phages have evolved inhibitors against McrBC and GmrSD[77,78], which confer the ability to bypass these defense machineries and allow phages to survive under conditions, in which they would normally be restricted. Knowing how these defense systems work, and how they have been naturally subverted is clinically important and will aid in the long-term development of small-molecule inhibitors that can impair conserved defense systems and improve the efficacy of phage-based treatments.

## Methods
**Cloning, expression, and purification of TgMcrB^AAA.** The gene for the *T. gammatolerans EJ3* McrB protein (JGI IMG/M ID 644807740) was codon optimized for expression in *E. coli* and synthesized commercially by GENEART (Supplementary Table 1). The DNA for the AAA+ domain of TgMcrB (residues

186–613) was amplified by PCR and cloned via Gibson assembly (New England Biolabs) into the pET15bP vector, a modified version of the pET15b vector, in which the Factor Xa cleavage site after the N-terminal 6xHis tag was replaced with an HRV 3C cleavage site. Cleavage by HRV 3C protease leaves a glycine and a proline residue immediately upstream of TgMcrB$^{AAA}$'s N-terminal methionine. Primers used in this study are summarized in Supplementary Table 2.

Selenomethionine-labeled (SeMet) TgMcrB$^{AAA}$ was expressed in minimal medium using methionine auxotrophs (T7 Express Crystal Competent E. coli, New England Biolabs) according to manufacturer's protocols. For the expression of native TgMcrB$^{AAA}$, the construct was transformed into E. coli BL21(DE3) cells, which were grown at 37 °C in Terrific Broth. When OD600 reached 1.0, protein expression was induced by addition of 0.3 mM isopropyl β-D-thiogalactoside (IPTG) and cells were grown overnight at 19 °C. Cells were harvested by centrifugation at $6000 \times g$ for 15 min at 25 °C, and washed twice with nickel-loading buffer (NLB; 20 mM HEPES, pH 7.5, 500 mM NaCl, 30 mM imidazole, 5% glycerol (v/v), and 5 mM β-mercaptoethanol). Pellets were typically flash frozen in liquid nitrogen and stored at −80 °C.

Thawed pellets from 500-mL cultures were resuspended in 30 mL of NLB supplemented with 10 mM PMSF, 5 μg/mL DNase I (Roche), 5 mM MgCl₂, and a tablet of complete protease inhibitor cocktail (Roche). Lysozyme was added to a final concentration of 1 mg/mL and the mixture was incubated for 15 min at 4 °C with rocking. Cells were disrupted by sonication and the lysate was cleared of debris by centrifugation at $19,700 \times g$ for 30 min at 4 °C. The supernatant was filtered using a 0.45-μm cutoff syringe filter, incubated at 65 °C for 20 min, centrifuged at $6000 \times g$ for 15 min at 4 °C, and loaded onto a 5-mL HiTrap chelating column (GE Healthcare) charged with NiSO₄ and then washed with NLB. TgMcrB$^{AAA}$ was eluted with an imidazole gradient from 30 mM to 1 M. Peak fractions were pooled, HRV 3C protease was added, and the sample was dialyzed overnight at 4 °C against cleaning buffer (20 mM HEPES, pH 7.5, 50 mM NaCl, 5% glycerol (v/v), and 5 mM β-mercaptoethanol (10 mM for SeMet-labeled protein)). Another 5-mL HiTrap chelating column charged with NiSO₄ was equilibrated with cleaning buffer and the sample was applied to this column, followed by elution with a NLB to remove the cleaved 6xHis tag. Pooled peak fractions were concentrated to 2 mL with a centrifugal concentrator (50 kDa cutoff, Millipore). The concentrated protein was further purified by size-exclusion chromatography (SEC) using a HiLoad 16/600 Superdex 200 pg column (GE Healthcare). During SEC, all proteins were exchanged into SEC$_{150}$ buffer (20 mM HEPES, pH 7.5, 150 mM KCl, 5 mM MgCl₂, and 1 mM DTT (5 mM for SeMet TgMcrB$^{AAA}$)). For crystallographic analysis, the protein was concentrated to 40–80 mg/mL. Concentrations of purified proteins were determined by SDS–PAGE with BSA standards. All point mutations were introduced into TgMcrB$^{AAA}$ in the pET15bP vector by quick-change PCR and the proteins were purified as described above.

**Cloning, expression, and purification of TgMcrB.** The gene for full-length TgMcrB (residues 1–613) was amplified by PCR and cloned into the pET15bP vector via Gibson assembly. Cleavage by HRV 3C protease leaves a glycine and a proline residue immediately upstream of TgMcrB's N-terminal methionine. The construct was transformed into E. coli BL21(DE3) cells, which were grown at 37 °C in Terrific Broth. When OD600 reached 1.5, protein expression was induced with 0.3 mM IPTG and cells were grown overnight at 19 °C. Cells were harvested and washed twice with NLB. Pellets were typically flash frozen in liquid nitrogen and stored at −80 °C. Thawed pellets from 2-L cultures were resuspended in 30 mL of NLB supplemented with 10 mM PMSF, 5 μg/mL DNase I, 5 mM MgCl₂, and a tablet of complete protease inhibitor cocktail. Cells were lysed and the full-length TgMcrB protein was purified as described above with the slight modification of using 250 mM KCl in the SEC buffer (20 mM HEPES, pH 7.5, 250 mM KCl, 5 mM MgCl₂, and 1 mM DTT). The protein was concentrated to 20–40 mg/mL.

**Cloning, expression, and purification of TgMcrC.** The gene for the T. gamma-tolerans EJ3 McrC protein (JGI IMG/M ID 644807739) was codon optimized for expression in E. coli and synthesized commercially by Integrated DNA Technologies (Supplementary Table 1). The DNA encoding full-length TgMcrC (residues 1–458) was amplified by PCR and cloned via Gibson assembly into the pCAV6 vector, a modified version of the pMAL c5x T7 expression vector, in which a 6xHis tag was introduced upstream of the N-terminal MBP sequence and an HRV 3C cleavage site replaces that for Factor Xa in the multiple cloning site. Cleavage by HRV 3C protease leaves a glycine and a proline residue immediately upstream of TgMcrC's N-terminal methionine.

The TgMcrC construct was transformed into E. coli BL21(DE3) cells, which were grown at 37 °C in Terrific Broth. When OD600 reached 1.0, protein expression was induced with 0.3 mM IPTG and the cells were grown overnight at 19 °C. Cells were harvested and washed twice with NLB. Pellets were typically flash frozen in liquid nitrogen and stored at −80 °C. Thawed pellets from 500-mL cultures were resuspended in 30 mL of NLB supplemented with 10 mM PMSF, 5 μg/mL DNase I, 5 mM MgCl₂, and a tablet of complete protease inhibitor cocktail. Lysozyme was added to a final concentration of 1 mg/mL and the mixture was incubated for 15 min at 4 °C with rocking. Cells were disrupted by sonication and the lysate was cleared of debris by centrifugation at $19,700 \times g$ for 30 min at 4 °C. The supernatant was filtered using a 0.45-μm cutoff syringe filter, loaded onto a 5-mL HiTrap chelating column charged with NiSO₄ and then washed with NLB.

TgMcrC was eluted with an imidazole gradient from 30 mM to 1 M. Peak fractions were pooled, HRV 3C protease was added, and the sample was dialyzed overnight at 4 °C against SP-loading buffer (SPLB; 20 mM HEPES, pH 7.5, 250 mM NaCl, 1 mM EDTA, 5% glycerol (v/v), and 1 mM DTT). The sample was applied to a 5-mL HiTrap SP HP column (GE Healthcare) equilibrated with SPLB and then washed with SPLB. TgMcrC was eluted with a NaCl gradient from 250 mM to 1 M. Because TgMcrC is prone to precipitate, no further purification steps were attempted and the pooled peak fractions yielded protein at a purity of ~70% and a concentration of ~0.8 mg/mL. All point mutations were introduced into TgMcrC in the pCAV6 vector by quick-change PCR and the proteins were purified as described above.

**Cloning, expression, and purification of EcMcrB.** The gene for full-length E. coli McrB (Uniprot P15005; JGI IMG ID 646316336) was codon optimized for expression in E. coli and synthesized commercially by GENEART (Supplementary Table 1). The DNA encoding full-length EcMcrB (residues 1–459) was cloned into the pMAL-c2XP vector, a modified version of the pMAL-c2X vector (New England Biolabs), in which the Factor Xa cleavage site after the N-terminal MBP tag was replaced with an HRV 3C cleavage site. Cleavage by HRV 3C protease leaves a glycine and a proline residue immediately upstream of EcMcrB's N-terminal methionine.

EcMcrB was transformed into E. coli BL21(DE3) cells, which were grown at 37 °C in Terrific Broth. When OD600 reached 1.0, protein expression was induced with 0.3 mM IPTG, and cells were grown overnight at 19 °C. Cells were harvested and washed once with TGED$_{500}$ buffer (20 mM Tris-HCl, pH 8.0, 5% glycerol (v/v), 1 mM EDTA, 1 mM DTT, and 500 mM NaCl). Pellets were flash frozen in liquid nitrogen and stored at −80 °C. Thawed pellets from 500-mL cultures were resuspended in 30 mL of TGED$_{500}$ buffer supplemented with 10 mM PMSF, 5 μg/mL DNase I, 5 mM MgCl₂, and a tablet of complete protease inhibitor cocktail. Lysozyme was added to a final concentration of 1 mg/mL and the mixture was incubated for 15 min at 4 °C with rocking. Cells were disrupted by sonication and the lysate was cleared of debris by centrifugation at $19,700 \times g$ for 30 min at 4 °C. The supernatant was filtered using a 0.45-μm cutoff syringe filter, loaded onto a 5-mL HiTrap MBP column (GE Healthcare), washed with TGED$_{500}$, and eluted with 10 mM D-maltose in TGED$_{500}$ buffer. Peak fractions were pooled, HRV 3C protease was added, and the sample was dialyzed overnight at 4 °C against TGED$_{50}$ buffer (TGED$_{500}$ buffer but with 50 mM NaCl instead of 500 mM). The sample was then applied to a 5-mL HiTrap Q HP ion-exchange column (GE Healthcare) in TGED$_{50}$ and eluted with a NaCl gradient from 50 to 500 mM. Peak fractions were pooled, concentrated, and further purified by SEC using a HiLoad 16/600 Superdex 200 pg column, during which the protein was exchanged into SEC$_{150}$ buffer. The protein was then concentrated to ~25 mg/mL.

**Cloning, expression, and purification of EcMcrC.** The gene encoding full-length E. coli McrC protein (Uniprot P15006; JGI IMG ID 637004274) was codon optimized for expression in E. coli and synthesized commercially by GENEART (Supplementary Table 1). The DNA encoding full-length EcMcrC (residues 1–348) was cloned into the pMAL-c2XP vector. Cleavage by HRV 3C protease leaves a glycine and a proline residue immediately upstream of EcMcrC's N-terminal methionine.

EcMcrC was transformed into E. coli BL21(DE3) cells and grown at 37 °C in Terrific Broth. When OD600 reached 1.0, protein expression was induced with 0.3 mM IPTG and cells were grown overnight at 19 °C. Cells were harvested and washed once with TGED$_{500}$ buffer. Pellets were flash frozen in liquid nitrogen and stored at −80 °C. Thawed pellets from 500-mL cultures were resuspended in 30 mL of TGED$_{500}$ buffer supplemented with 10 mM PMSF, 5 μg/mL DNase I, 5 mM MgCl₂, and a tablet of complete protease inhibitor cocktail. Lysozyme was added to a final concentration of 1 mg/mL and the mixture was incubated for 15 min at 4 °C with rocking. Cells were disrupted by sonication and the lysate was cleared of debris by centrifugation at $19,700 \times g$ for 30 min at 4 °C. The supernatant was filtered using a 0.45-μm cutoff syringe filter, loaded onto a 5-mL HiTrap MBP column, washed with TGED$_{500}$ buffer, and eluted with 10 mM D-maltose in TGED$_{500}$ buffer. Peak fractions were pooled, HRV 3C protease was added, and the sample was dialyzed overnight at 4 °C against HGED$_{250}$ buffer (20 mM HEPES, pH 7.5, 5% glycerol (v/v), 1 mM EDTA, 1 mM DTT, and 250 mM NaCl). The sample was then applied to a 5-mL HiTrap SP HP ion-exchange column in TGED$_{250}$ buffer and eluted with a NaCl gradient from 250 mM to 1 M. Because EcMcrC is prone to precipitate, no further purification steps were attempted. The pooled peak fractions yielded protein at a purity of ~70% and a concentration of ~6 mg/mL.

**GTPase activity assays.** GTPase activity was measured by using a colorimetric malachite green assay that monitors the amount of free phosphate released over time[79]. To measure the basal GTPase activity of TgMcrB$^{AAA}$, 0.4 μM TgMcrB$^{AAA}$ was incubated with 1 mM GTP at 65 °C in reaction buffer (20 mM Tris-HCl, pH 8.0, 150 mM KCl, and 5 mM MgCl₂). To measure the GTPase activity of TgMcrB$^{AAA}$ stimulated by TgMcrC, the same conditions were used but 0.1 μM TgMcrC was added. At time points of 0, 5, 10, 20, 30, 45, 60, 80, 100, and 120 min, 20-μL aliquots were taken and quenched with 5 μL of 0.5 M EDTA, pH 8.0. For colorimetric reactions, 150 μL of filtered malachite green solution were added to

each sample and incubated for 5 min. The absorbance at 650 nm of the samples was measured with a Multiskan GO Microplate Spectrophotometer (Thermo Scientific). The amount of phosphate released was determined using a standard curve. To account for the spontaneous hydrolysis of GTP at 65 °C, a protein-free sample containing GTP and magnesium was incubated in parallel, and the measured amount of phosphate released at each time point was subtracted from the corresponding measurements of protein-containing samples. The specific activity is reported for all wild-type and mutant proteins. Quantified data represent the average of three independent experiments using multiple independently purified batches of protein with error bars indicating the standard deviation from the mean ($n = 3$, mean ± standard deviation). GraphPad Prism and Microsoft Excel were used for statistical analysis and to plot the data.

**Negative-stain EM.** Negatively stained samples were prepared as described[80]. Freshly purified proteins were diluted to ~0.05 mg/mL with SEC150 buffer supplemented with 2.5 mM GTPγS before applying 5-µL aliquots to glow-discharged grids. Grids were stained with 0.7% uranyl formate (Pfaltz & Bauer, U01000) and imaged with a Philips CM10 electron microscope equipped with a tungsten filament and operating at 100 kV. All images were recorded on an AMT XR16L-ActiveVu charge-coupled device camera (Woburn, MA, USA) using a defocus of approximately −1.5 µm and a nominal magnification of 52,000×.

**Crystallization, X-ray data collection, and structure determination of TgMcrB^AAA.** SeMet TgMcrB^AAA was diluted to 16 mg/mL with SEC150 buffer containing 2.5 mM GTPγS and crystallized by sitting drop vapor diffusion in 0.1 M sodium acetate, pH 6.5, 17.5% 2-methyl-2,4-pentanediol (v/v) with a drop size of 2 µL, and a reservoir volume of 650 µL. Crystals appeared within 3–4 days at 20 °C and were cryo-protected with Parabar 10312 (Hampton Research) and frozen in liquid nitrogen. Single-wavelength anomalous diffraction (SAD) data were collected remotely on the tunable NE-CAT 24-ID-C beamline at the Advanced Photon Source at the selenium edge energy of 12.663 keV (0.9791 Å; Supplementary Table 3). Data were integrated and scaled using the NE-CAT RAPD pipeline, which utilizes LABELIT[81], RADDOSE[82], BEST[83], MOSFLM[84], Xtriage from PHENIX[85], XDS[86], and AIMLESS[87]. Strong anomalous signal was obtained from a single crystal diffracting to 2.9 Å (space group $P2_1$; unit cell dimensions: $a = 100.24$ Å, $b = 108.87$ Å, $c = 118.67$ Å and $\alpha = 90.00°$, $\beta = 107.41°$, $\gamma = 90.00°$), and multiple SeMet SAD datasets were collected from different positions of this crystal. All possible combinations of datasets were tested and merged using the program BLEND in the CCP4 suite[88,89]. Experimental phases were obtained from the combination with the strongest anomalous signal. Heavy-atom sites were located using SHELX C/D/E[90] in the CCP4 suite and phasing, density modification, and initial model building were carried out using the CRANK2 pipeline[91] in the CCP4 suite. Iterative rounds of refinement and model building were carried out using the programs COOT[92] and REFMAC[93] in the CCP4 suite to improve the initial model, which resolved most regions of four TgMcrB^AAA monomers. Native TgMcrB^AAA with 2.5 mM GTPγS was crystallized at a concentration of 16 mg/mL by sitting drop vapor diffusion in 0.1 M sodium acetate, pH 6.5, 16.75% (v/v) 2-methyl-2,4-pentanediol with a drop size of 2 µL, and a reservoir volume of 650 µL. Crystals were frozen as described above. The partial model from the SeMet SAD datasets was used as the search model to perform molecular replacement, using PHASER[94] in the CCP4 suite on a native dataset diffracting to 2.83 Å (space group $P2_1$; unit cell dimensions: $a = 100.02$ Å, $b = 108.55$ Å, $c = 118.43$ Å and $\alpha = 90.00°$, $\beta = 106.94°$, $\gamma = 90.00°$), which was collected at the NE-CAT 24-ID-E beamline at the selenium edge energy of 12.663 keV (0.9791 Å). Further model building and refinement was carried out manually in COOT and PHENIX, respectively[85,92]. Non-crystallographic symmetry was enforced during the refinement with no additional constraints imposed. The final model contained four well-resolved and two poorly resolved molecules in the asymmetric unit and was refined to 2.95 Å resolution with $R_{work}/R_{free}$ values of 0.345/0.364 (Supplementary Table 3). Poorly resolved electron density surrounding the small subdomain of subunit F—near where each hexamer contacts its neighbor in the crystal lattice—and our inability to properly build and refine this portion of the model both contributed to the high $R$ values.

**Cryo-EM sample preparation and data collection.** For TgMcrB^AAA, thawed protein was diluted to 10 mg/mL with SEC150 buffer containing 2.5 mM GTPγS. Samples were mixed with 20× digitonin (Calbiochem) stock to a final concentration of 0.05%, and 3.5 µL aliquots were applied to C-flat thick holey carbon grids (CF-1.2/1.3-4C, Protochips), blotted for 7 s at 4 °C and plunge-frozen in liquid ethane using a Vitrobot Mark IV (Thermo Fisher Scientific).

For the TgMcrB^AAAC complex, thawed TgMcrB^AAA was mixed with freshly purified TgMcrC at a molar ratio of 4:1. The sample was concentrated using a 2-mL centrifugal concentrator (100 kDa cutoff, Millipore). Concentrated protein was buffer-exchanged to SEC150 buffer in the concentrator, and the final concentration was estimated to be ~14 mg/mL by measuring the absorbance at 280 nm. The complex was then mixed with 50× GTPγS stock solution to a final GTPγS concentration of 2.5 mM and incubated for 30 min at 4 °C. Samples were mixed with 20× digitonin to a final concentration of 0.05%, and 3.5-µL aliquots were

applied to C-flat thick holey carbon grids (CF-1.2/1.3-4C-T), blotted for 8–10 s at 4 °C and plunge-frozen in liquid ethane, using a Vitrobot Mark IV.

For the TgMcrBC and EcMcrBC complexes, thawed McrB was mixed with freshly purified McrC at a molar ratio of 4:1. The samples were concentrated using 2-mL centrifugal concentrators (100 kDa cutoff, Millipore). Concentrated proteins were buffer-exchanged into SEC250 buffer (for TgMcrBC) or SEC150 buffer (for EcMcrBC) in the concentrators and finally concentrated to ~16 mg/mL. The prepared complexes were mixed with 50× GTPγS stock solution to a final GTPγS concentration of 2.5 mM and incubated for 30 min at 4 °C. Samples were mixed with 20× digitonin to a final concentration of 0.05% digitonin, and 3.5 µL aliquots were applied to Quantifoil R1.2/1.3 400 mesh Au grids, blotted for 8–10 s at 4 °C, and plunge-frozen in liquid ethane, using a Vitrobot Mark IV.

Cryo-EM data were collected on a 300-kV Titan Krios electron microscope (Thermo Fisher Scientific) equipped with a K2 Summit direct electron detector at a nominal magnification of 29,000× in super-resolution counting mode using SerialEM[95,96]. After binning over 2 × 2 pixels, the calibrated pixel size was 1.0 Å on the specimen level. For all specimens other than EcMcrBC, exposures of 10 s were dose-fractionated into 40 frames with a dose rate of 8 electrons per pixel per second, resulting in a total dose of 80 electrons per Å². For EcMcrBC, exposures of 20 s were dose-fractionated into 40 frames with a dose rate of 4 electrons per pixel per second, resulting in a total dose of 80 electrons per Å². Cryo-EM data collection statistics are summarized in Supplementary Table 4.

**Cryo-EM data processing.** For TgMcrB^AAA and the TgMcrB^AAAC complex, image processing was done in RELION-3.0-beta[97–99], and images for TgMcrBC and EcMcrBC were processed in both CryoSPARC-2.4.0 (Structura Biotechnology)[100] and RELION-3.0. All movie frames were corrected with a gain reference collected during the same EM session, and specimen movement was corrected using RELION's implementation of motion correction (for TgMcrB^AAA and TgMcrB^AAAC) or MotionCorr2 (for TgMcrBC and EcMcrBC) with dose weighting[97,101]. The contrast transfer function (CTF) parameters were estimated using CTFFIND-4.1.8 (ref. [102]) for TgMcrB^AAA and TgMcrB^AAAC or Gctf-1.0.6 (ref. [103]) for TgMcrBC and EcMcrBC. Images showing substantial ice contamination, abnormal background, thick ice, low contrast, or poor Thon rings were discarded.

For TgMcrB^AAA, 1599 micrographs were collected, of which 1517 micrographs were selected for further processing. Particles were picked with Gautomatch (https://www.mrc-lmb.cam.ac.uk/kzhang/Gautomatch/) without templates, which identified 277,503 particles that were windowed into 320 × 320 pixel images. The particle images were binned four times and subjected to two rounds of 2D classification. Classes that produced averages with fine structural detail and showed no overlap with neighboring particles were combined and used to generate an initial reference map. The selected 153,891 particles were subjected to 3D classification into four classes, three of which were selected and used to re-extract the corresponding particles into 320 × 320 pixel images that were then rescaled into 256 × 256 pixel images. The centered, re-extracted particles were refined with C1 symmetry to a resolution of 3.4 Å according to the Fourier shell correlation (FSC) = 0.143 criterion[104], which was used for all resolution estimates. Subsequent CTF refinement and Bayesian polishing improved the overall resolution of the map to 3.1 Å.

For the TgMcrB^AAAC complex, 1795 of the 2070 collected micrographs were selected for further processing. Gautomatch was used to pick the first 200 micrographs without templates, and then ~10,000 picked particles were subjected to 2D classification. Four representative class averages were then selected as templates for Gautomatch to pick particles from all the micrographs. The 264,850 auto-picked particles were cleaned-up by two rounds of 2D classification. The particles from eight classes with well-defined averages (156,149 particles) were used to generate an initial density map in RELION, which was then used as reference for 3D classification of the cleaned-up particles into six classes. Four classes showed good fine structure and were combined (115,774 particles), and subsequent 3D refinement with C1 symmetry, CTF refinement, and Bayesian polishing yielded a map at 4.4-Å resolution. A second dataset collected using the same conditions was processed following the same strategy, yielding a map at 4.3-Å resolution from 88,819 refined particles. The particles from the two datasets were combined and further refined with C1 symmetry to generate an improved map at 4.2 Å (204,593 particles). While this map showed strong density for one half of the complex, the other half was represented by substantially weaker density. 3D refinement was thus repeated with C2 symmetry imposed, which yielded a symmetrized map for the full complex map at 4.2-Å resolution. To overcome the flexibility of the connection between the two half-complexes, particles in the nonsymmetrized map were subjected to automated multibody refinement implemented in RELION-3, using individual masks for the two half-complexes that overlapped in the region of the twofold axis. Signal subtraction was performed for each rigid body using the "relion_flex_analyse" command, which only retains the signal inside the selected rigid body[99]. The signal-subtracted particles for one of the two bodies were used to calculate a reference map for the half-complex, using the "relion_reconstruct" command. The signal-subtracted particles for both bodies were combined (409,186 particles) and subjected to 3D refinement with C1 symmetry and starting with a global search. Subsequent CTF refinement and 3D refinement yielded the final map for the half-complex at an overall resolution of 3.7 Å.

For the TgMcrBC complex, 1936 of 2078 micrographs and for the EcMcrBC complex, 1088 of 1161 micrographs were selected for further processing. Particles were picked with Gautomatch with templates generated from preliminary data collected on a 200-kV Talos Arctica electron microscope (Thermo Fisher Scientific). The auto-picked particles (354,707 for the TgMcrBC complex and 184,487 for the EcMcrBC complex) were extracted into $320 \times 320$ pixel images that were then rescaled into $256 \times 256$ pixel images. All particle images were used for ab initio reconstruction in Cryosparc-2.4.0, specifying three output classes. The best of the three maps, including 226,813 particles for the TgMcrBC complex and 106,684 particles for the EcMcrBC complex, were selected for nonuniform refinement, which yielded maps for the half-complexes at 3.0-Å resolution for the TgMcrBC complex and at 4.9-Å resolution for the EcMcrBC complex. The particles were transferred back to RELION using the pyem package (https://github.com/asarnow/pyem), re-extracted into $320 \times 320$ pixel images, and further refined without imposing symmetry to generate maps for the TgMcrBC complex at 2.9-Å resolution and for the EcMcrBC complex at 4.1-Å resolution. CTF refinement and Bayesian polishing improved the maps to resolutions of 2.7 and 3.5 Å, respectively. Finally, the particles were re-extracted into $400 \times 400$ pixel images. Refinement, CTF refinement and Bayesian polishing yielded the final maps at 2.4-Å resolution for TgMcrBC and at 3.3-Å resolution for EcMcrBC.

**Model building and refinement.** For the TgMcrB[AAA] hexamer, the best refined monomer from the X-ray model was used and fit into each subunit density of the 3.1-Å resolution cryo-EM map using UCSF Chimera[105]; no other information from the X-ray structure was used. Further iterative refinement cycles between the phenix.real_space_refine command in PHENIX with secondary structure restraints and manual adjustments in COOT yielded the final model for the TgMcrB[AAA] hexamer.

For the TgMcrB[AAA]C complex, the final cryo-EM model of the TgMcrB[AAA] hexamer was manually fit into the 3.7-Å resolution cryo-EM map of the half-complex and refined using the phenix.real_space_refine command in PHENIX with morphing, simulated annealing, and secondary structure restraints. Ab initio model building for TgMcrC was carried out in COOT[92], guided by secondary structure predictions from SPIDER2[106] and PSIPRED[107]. The density for TgMcrC was good up to residue 312, but the remaining C-terminal endonuclease domain was poorly resolved. Therefore, a homology search was performed in I-TASSER[108–110] for TgMcrC residues 312–458, and a homology model was generated based on the *Saccharolobus solfataricus* Holliday junction resolving enzyme[57] (PDB: 1OB8; sequence identity: 17%). This homology model was fit into the corresponding density and manual adjustments were performed in COOT. Finally, all built models were combined and iterative cycles of real-space refinement in PHENIX with secondary structure restraints and manual adjustments in COOT were performed, yielding the final model for the TgMcrB[AAA]C half-complex.

For the TgMcrBC complex, the final cryo-EM model of TgMcrB[AAA]C was manually fit into the 2.4-Å cryo-EM map of the half-complex and refined using the phenix.real_space_refine command in PHENIX with morphing, simulated annealing, and secondary structure restraints. Because the nuclease domain of TgMcrC was poorly resolved, most regions were removed from the model. Finally, iterative cycles of real-space refinement in PHENIX with secondary structure restraints and manual adjustments in COOT were performed, yielding the final model of the TgMcrBC half-complex.

For the EcMcrBC complex, SWISS-MODEL[111] was used to generate a homology model of EcMcrB based on the TgMcrB[AAA] structure (sequence identity: 20% across the whole protein, 24% across the AAA+ domain alone), which was manually fit into one subunit in the 3.3-Å resolution cryo-EM map using UCSF Chimera. After manual adjustments in COOT, the corrected model was fit into each subunit of the hexameric EcMcrB density. To build the EcMcrC model, the unique N-terminal domain was removed from the TgMcrC model, and all the residues were mutated to alanine except for the highly conserved residues (based on the sequence alignment of McrC homologs; sequence identity: 13% across the whole protein, 20% across the finger domain alone). This model was manually fit into the corresponding density of the 3.3-Å resolution cryo-EM map in UCSF Chimera, followed by manual adjustment of each residue in COOT. Manual adjustment was guided by secondary structure predictions from SPIDER2 and PSIPRED. Finally, all built models were combined, and iterative cycles of real-space refinement in PHENIX with secondary structure restraints and manual adjustments in COOT yielded the final model of the EcMcrBC half-complex.

All the refinement statistics are summarized in Supplementary Table 4. For model validation, the final model for each map was refined against one of the independent half maps (map 1) of the corresponding map. FSC curves were then calculated between the refined model and half map 1 (work), half map 2 (free), as well as the combined map (Supplementary Figs. 1h, 4f, 7f and 10g).

**Statistics and reproducibility.** Several cryo-EM data collection sessions including ones for grid screening were performed for each protein complex, but only the best datasets (usually one dataset for each structure except for the TgMcrB[AAA]C complex) are reported and were processed as described in "Methods" section. The freezing conditions of cryo-EM grids were always duplicated or triplicated with a small variation at the blotting times (7–10 s), providing us reproducible micrograph images and more chances to yield the best ice thickness over one grid for each data collection. The cryo-EM images shown in Supplementary Figs. 1b, 4a and 7a are representative micrographs in >1000 similar micrographs. The 2D-class averages shown in Supplementary Figs. 1c, 4b, 7b and 10a represent eight classes with the most particles. The negative-stain EM images shown in Fig. 2e are representative micrographs from the reproducible EM data taken every time together with the three independent experiments for GTPase activity assays.

**Reporting summary.** Further information on research design is available in the Nature Research Reporting Summary linked to this article.

## Data availability

Protein sequences are available from either Uniprot (https://www.uniprot.org/) or the Joint Genome Institute Integrated Microbial Genomes and Microbiomes (JGI IMG/M) database (https://img.jgi.doe.gov/). Atomic coordinates and structure factors for TgMcrB[AAA] have been deposited in the Protein Data Bank (https://www.rcsb.org/) with the accession code 6UT3. The B-factor sharpened 3D cryo-EM density maps and atomic coordinates of EcMcrBC, TgMcrB[AAA], TgMcrBC, TgMcrB[AAA]C (full mask), and TgMcrB[AAA]C (combined) have been deposited in the Worldwide Protein Data Bank (http://www.wwpdb.org/) under accession numbers EMD-20867 and 6UT6, EMD-20865 and 6UT4, EMD-20866 and 6UT5, EMD-20868 and 6UT7, EMD-20871 and 6UT8, respectively. The B-factor sharpened 3D cryo-EM density maps of the two multibody-refined body 1 and body 2 have been deposited in the Worldwide Protein Data Bank (wwPDB) under accession number EMD-20869 (body 1) and EMD-20870 (body 2), respectively. Other data are available from the corresponding authors upon reasonable request. Source data are provided with this paper.

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

## Acknowledgements

We are grateful to Mark Ebrahim and Johanna Sotiris for help with grid screening and data collection at the Evelyn Gruss Lipper Cryo-Electron Microscopy Resource Center at The Rockefeller University. We thank Drs. Richard Cerione and Holger Sondermann for critical reading of the manuscript and the Northeastern Collaborative Access Team (NE-CAT) beamline staff at the Advanced Photon Source (APS) for assistance with remote X-ray data collection. This work was supported by National Institutes of Health Grant GM120242 (to J.S.C.) and is based upon research conducted at NE-CAT beamlines (24-ID-C and 24-ID-E) under the general user proposals GUP-51113 and GUP-41829 (PI: J.S.C.). NE-CAT beamlines are funded by the National Institute of General Medical Sciences from the National Institutes of Health (P30 GM124165). The Pilatus 6 M detector on 24-ID-C beamline is funded by a NIH-ORIP HEI grant (S10 RR029205). This research used resources of the Advanced Photon Source, a U.S. Department of Energy (DOE) Office of Science User Facility operated for the DOE Office of Science by Argonne National Laboratory under Contract No. DE-AC02-06CH11357. J.S.C. is a Meinig Family Investigator in the Life Sciences.

## Author contributions

Y.N., H.S., C.J.H., T.W., and J.S.C. designed the study and analysed data. C.J.H. and Y. N. cloned purified all constructs and carried out biochemical assays. Y.N. and H.S. collected cryo-EM data, carried out image processing, and built the atomic-resolution cryo-EM models. C.J.H. and Y.N. collected X-ray diffraction data, determined the X-ray structure, and built the X-ray model. Y.N., H.S., T.W., and J.S.C. wrote the manuscript.

## Competing interests

The authors declare no competing interests.
