## [Peer Review File · Nature Communications]

Reviewers' comments:

Reviewer #1 (Remarks to the Author):

Despite the fact that McrBC was among the first discovered methyl-directed restriction enzymes, its reaction mechanism remains elusive. Biochemical and EM studies conducted 20-30 years ago suggested that *E. coli* McrBC (EcMcrBC) follows a reaction mechanism reminiscent of Type I restriction enzymes, where NTP hydrolysis powers DNA translocation with concomitant loop extrusion. Recently, Nirwan et al. presented a cryo-EM structure of the EcMcrBC assembly, which consists of two hexameric McrB rings bridged by an McrC dimer.

In the present work, authors present a higher resolution cryo-EM structure of EcMcrBC, and even higher resolution cryo-EM and X-ray structures of McrB subcomplex and McrBC assembly of an McrBC homolog from *Thermococcus gammatolerans* (TgMcrBC). These structures provide new insights into the GTP (vs ATP) specificity of McrBC enzymes and indicate that McrB homohexamer has intrinsic asymmetry. Of particular interest is the proposed mechanism of GTP hydrolysis stimulation by McrC, which provides a positively charged "finger domain" residue (R263 in TgMcrBC, K157 in EcMcrBC) to the interface between two McrB subunits, thereby promoting optimal positioning of catalytic residues and ensuring that McrC stimulation at any given time occurs at a single active site.

The manuscript is clearly written, provided results are a valuable addition to the vast field of AAA+ ATPases, and thus this work may be of interest to a broad audience.

I have two questions that could be further discussed in this publication:

1. Authors show that the McrB hexamer even in the absence of McrC is asymmetric, with 4 "tight" and 2 "loose" inter-McrB interfaces. But what are the physical constraints of the McrB subunits that result in such obligatory asymmetry? Could it be that the angle between each pair of McrB subunits forming "tight" interfaces is less than 60° (say 59°); to fill the full 360° circle, the angle between the last McrB subunit and adjacent subunits must be greater than 60°, resulting in 2 "loose" interfaces?

2. Based on the proposed role of the McrC "finger" domain in GTP hydrolysis, and the observed interactions between McrB and McrC subunits, authors of the present work rule out the possibility of DNA threading through the central (McrB)₆ pore (authors of the previous cryo-EM study of EcMcrBC, Nirwan et al. 2019, considered such threading of both DNA and McrC through the McrB ring as a possibility). But if DNA is not threaded through the central channels of McrB hexamers, and both McrB rings rotate relative to the (McrC)₂ stalk, it gives us the following mechanism: the two-site DNA is bound by the McrBC tetradecamer, with each site interacting with DNA recognition domain of a different (McrB)₆ hexamer; ensuing rotation of DNA-bound McrB rings relative to McrC would wrap the intervening DNA around the McrBC complex (the dumbbell-shaped McrBC complex would act as a reel); ensuing DNA tension may position it in the catalytic cleft of (McrC)₂, resulting in DNA cleavage. Is such mechanism compatible with available biochemical data?

Minor points:

Fig. S1j - the hexamer opening is indicated for "Homodimer 1" only; it could be shown for "Homodimer 2" as well.

page 9 line 18 - "H₂O catalytic", should be "H₂O catalytic".

Based on the dimerisation mode of McrC, could the DNA stagger (length of 5'- or 3'-overhang) of the cleavage product be predicted?

1st paragraph of the "Introduction" and the last paragraph of the "Discussion" look somewhat artificial in the otherwise fluent story.

Giedrius Sasnauskas

Reviewer #2 (Remarks to the Author):

In this manuscript, Niu et al. Present cryo-EM structures of the McrB hexamer and McrBC complexes from both *Thermococcus gammatolerans* (Tg) and *Escherichia coli* (Ec). McrBC is a two-component modification-dependent restrictions system (MDRS) that restricts phage DNA and foreign DNA containing methylated cytosines. Elucidating the structure and function of bacterial defense systems, such as MDRS, is not only important on a fundamental level, but may aid in the development of new drugs that increase efficacy of phage therapy. While a structure of the AAA+ domain of McrB in complex with McrC was published previously, Niu et al. now provide structures with the full-length Ec and Tg McrB in complex with McrC, and also demonstrate that unbound McrB forms an asymmetric hexamer, which has important implications in the mechanism of MrcB activation by McrC. By comparing structures of Ec and Tg McrBC, the authors show that the two McrBC complexes use different DNA-binding domains that use a general GTP-recognition mechanism employed by all G proteins. Finally, based on the asymmetry present in unbound MrcB and the McrBC complex, and the observed differences in the consensus loop present at the active sites of the McrB hexamer, the authors propose a coordinated and directional GTP-hydrolysis cycle for McrBC.

Major remarks:

1) A previous paper is cited (Nirwan et al., Nature Communications, 2019) that had previously characterized a complex between truncated *E. coli* MrcB and full-length MrcC. While the authors do cite the paper of Nirwan et al., it is not discussed in detail at all, and none of the available structures of the *E. coli* McrBC complex with three bound GNP and three bound GDP molecules are compared with the GTP γ S and GDP bound structures presented here. While I agree that the paper of Niu et al. provides important new mechanistic insights, I cannot support publication in Nature Communications until the authors explicitly and thoroughly compare their work with the previously published study (see also below), and clearly state the similarities and differences between the structures from the work of Nirwan et al. and the manuscript by Niu et al. presented here.

2) For dataset processed to 2.95 Angstrom with fairly decent statistics, the R/Rfree is excessively high (34.5/36.4). Why is that? Did the authors observe any crystal pathologies such as twinning or anisotropy? Maybe the data could be processed to a lower space group to see if R/Rfree improves and density for the two poorly-resolved molecules improves?

Minor remarks (all related to the major remark above):

1) In the introduction, the authors cite the paper by Nirwan et al., which provided a first structural view of McrBC by cryo-EM. They state that 'this work fell short of answering many important mechanistic questions'. It would be crucial if the authors elaborated more on the main findings of the manuscript by Nirwan et al., and on which questions remained unanswered.

2) Page 7, lines 5 – 6: The authors write that 'Trp223, which lies adjacent to the Walker A motif, forms a unique parallel π -stacking interaction from below that has never been observed nor predicted for any GTPase or AAA+ protein' and that 'Interestingly, the analogous residue (Phe209) was never mutated in previous studies of EcMcrB as it is not strictly conserved across the McrB family'. Based on what the authors observe in their TgMcrB structure, I find it odd that they do not compare their structure to the previously published one of EcMcrB, which clearly has a π -stacking interaction between Phe209 and the guanine base of GTP γ S (see figure 1, panel g of the paper by Nirwan et al., 2019).

3) Page 10, lines 20-21: The authors start the paragraph with 'To establish whether different homologs use a conserved mechanism for stimulated hydrolysis, we determined the single-particle

cryo-EM structure of the complex formed by the full-length E. coli proteins in the presence of GTPγS'. I understand that the work on E. coli MrcBC was most probably performed before or during the time the manuscript of Nirwan et al. was published. However, since this happened already in July 2019, the authors cannot pretend this manuscript and the structure described therein do not exist. They should at least compare their structure with the already published one.

4) On page 11 lines 28 - 30 the authors write 'While we could not calculate a reconstruction for the full EcMcrBC tetradecameric assembly, the map for the half-complex contains density for an additional ordered nuclease domain of EcMcrC (Figure S5f)'. Yet a reconstruction of the EcMcrBAAA+C tetradecameric assembly is readily available (Nirwan et al., 2019). Why is this information not used to compare with the TgMcrBAAA+C tetradecameric structure?

5) As written on page 13, and shown in figure S7, the authors perform modeling of a MrcBC:DNA complex based on a structural comparison with EndoMS. A similar comparison is performed in the paper by Nirwan et al. 2019. (Supplementary figure 13). Similar to what is discussed in Niu et al., Nirwan et al. observe that: 'a particularly interesting feature of McrBΔNC is that the McrC stalk blocks the pore of the McrB ring.' They state that 'It is possible that the DNA substrate requires sliding in through the widest interface cleft (the FA interface in the structure) or that the enzyme complex disassembles and reassembles on the substrate.' Why is the latter hypothesis (i.e. complex disassembly and reassembly on the substrate) not discussed by Niu and colleagues? Furthermore, Nirwan et al speculate that 'Like many other AAA+ proteins bound to their DNA substrate, rearrangements at the GDP-bound interface could open the McrB hexamer to form a spiral/lockwasher structure and accommodate the DNA.' This 'open spiral' or 'lockwasher' structure of MrcB corresponds to the MrcBAAA+ crystal structure one observed by Niu et al. Could it be that the open spiral state is more than just an artefact imposed by crystal packing?

6) In the discussion, the authors propose a sequential mechanism for GTP hydrolysis by MrcBC. How does the mechanism compare to the one proposed by Nirwan et al.,?

Small comments:

Page 3, line 28: 'Stimulation of hydrolysis by a binding partner is also rare among AAA+ proteins'. The E. coli MoxR AAA+ ATPase RavA is actually a nice example of this: binding by the inducible lysine decarboxylase LdcI stimulates RavA activity at low pH (Jessop et al., Commun. Biol., 2020).

Reviewer #3 (Remarks to the Author):

see attachment

March 22, 2020

The manuscript describes cryoEM structures of the AAA+ domain hexamer of the 5-methylcytosine-specific restriction enzyme complex B (McrB) from *Thermococcus gammatolerans* with and without the McrC subunit, and of both the *Th.gammatolerans* and *E.coli* structures of the full length McrBC complexes, in addition to the X-ray structure of the AAA domain hexamer of *Th.gammatolerans* McrB. The structures are very interesting and should be of interest to a wide readership, thus I recommend publication in *Nature Communications*, following a minor revision to address the points raised in the following.

An earlier paper (Nirwan et al, 2019b) also described cryoEM structures of the AAA domain hexamer of *E.coli* McrB in complex with McrC and already found the asymmetric interaction of the stalk of McrC with the distorted McrB-hexamer, which lead the authors to the hypothesis of a consecutive GTP hydrolysis cycle reminiscent of the F1-ATPase. However, the current manuscript adds many structural and analysis details, and the comparison between the *E.coli* and the new *Th.gammatolerans* structures is interesting and merits publication. The presentation is already very nice and easy to follow, but could probably be slightly improved by adding some more comparisons to the previous structure/publication by Nirwan et al (2019b in the current manuscript) and perhaps some additional figures as detailed below.

The analysis of the X-ray structure is very short, also the description of the corresponding X-ray methods (see also below). The overall temperature factor is very high which probably also (together with the disordered parts of the structure) explains the quite high R factors. Fig. S1j (that shows the distorted hexamer) is difficult to interpret since only one orientation is shown, it would be helpful to add e.g. a 90° rotated view. In the given orientation, it does not look like the symmetry-related molecule "pushes" one molecule out of the ring. Is it correct that the ring conformation looks like a snail/spiral? Is the density sufficient to identify the bound nucleotides and/or side chains in the interfaces? Crystal packing forces are usually weak, so the structure might tell something about the mechanism of ring opening/assembly... Are the GTPase-accelerating side chain interactions that are described for the cryoEM structures still possible in the distorted ring? Would one monomer fall off if the hydrolysis would proceed through the hexamer? Or is opening small enough to keep the interactions? A figure with a superimposition of the Xray/EM rings might be very helpful.

The information about which nucleotide is bound to which interface is apparently incomplete, at least I could not find this for all structures/interfaces. In addition, it would be nice to know how ambiguous/unambiguous the densities for each nucleotide are. Please compare those findings in more detail to Nirwan et al 2019b who found three GNP and three GDP nucleotides bound to the hexamer; the current manuscript apparently has e.g. five GTP- -S and one GDP (is that really GDP ?) bound to TgMcrBC (Fig. 4e-j).

It would of course be extremely interesting to see DNA bound to the tetradecamer. Did you also try to get the DNA complex? If yes, did the double-hexamer fall apart in presence of DNA or was just no DNA bound (similar to the Nirwan structure)? Even a negative stain structure would be very helpful...

It is evidently very difficult to decide from the current data which of the two main models is correct: The DNA either threads through the pore or binds on the outsides of the hexameric rings. But it seems to me that at least some more conclusions could be drawn from the structures. When I superimposed the Nirwan structures with the EndoMS (5GKF), it seemed like the DNA could be accommodated between the two nuclease domains that bridge the two hexameric rings with relatively modest hinge-bending motions of the McrC moiety as detailed below. (Nirwan 2019b described only the McrC dimer interface as a hinge, is there really only this one hinge possible?) A model with a DNA helix running from the nuclease domains out to the DNA binding domains could then reveal if the base pair distance between the cleavage site and the recognition motif would fit to the biochemical data. It would also be interesting to model the movement of the second hexamer assuming the first one is fixed (by its DNA binding domains) and the McrC stalk would rotate. Could this motion be somehow related to the DNA translocation which also requires GTP hydrolysis?

The full length structures with the DNA binding domains that are in different positions in EcMcrBC and TgMcrBC (here with a very short linker) are also very interesting, and they would be emphasized by a model that includes the DNA.

minor points (text inside stars is recommended to add to the manuscript):

page 3 line 30: "..is that *the second component* McrC only exerts..."

page 4 line 1: "..A recent cryo-EM reconstitution *at 3.6Å (PDB: 6HZ4, 6HZ7 etc)* of the hexameric..."

page 5 line 11-15: Is there any structural explanation for this (surprising) result that the mutations increase the hydrolysis rate?

page 5 line 16-24: Description of the X-ray structure is very short, please elaborate, see above. It would also be helpful to add after line 24 that for the solution/modelling of the cryoEM structures only the best defined monomer was used, but no other information from the X-ray structure (if that is correct).

page 6 line 24: "..than the Glu357Ala mutation *that leads to partial dissociation of the hexamers, probably due to the loss of the magnesium ions and subsequently lower nucleotide affinities* (Figure 2e)."

page 6 line 25: What *are* the "different effects on oligomerization .. Asp279 and Glu280 were mutated.." in EcMcrB (Nirwan, 2019a)? Therein, it was found that D279A/Q prevent oligomerization and do not bind nucleotide, whereas E280A/Q was still hexameric, but catalysis-impaired, which is quite consistent with your observations?

page 7 line 1: π -stacking of guanine base with Trp/Phe "..has never been observed..": It seems to be infrequent indeed, but e.g. in the YCJX (6NZ4) stress protein of Shewanella

oneidensis (Tsai et al, 2019) there is Trp110 stacking to the base (see Fig.A below). I did not check other residues besides Trp or all AAA structures... Please check those structures and phrase this statement more carefully.

Figure A: Guanine base stacking to Phe in EcMcrC superimposed with YcjX (6NZ4):

page 8 line 1: Did you use the EcMcrC to help with the sequence assignment? What is the sequence identity between EcMcrC and TgMcrC? And/or was the density sufficiently well-defined to allow unambiguous assignment in all parts of the McrC?

page 8 line 34: The 0.75Å overall RMSD between the hexameric rings TgMcrB^{AAA} and TgMcrBC is *extremely* small, please state exactly for which atoms this value was calculated (backbone atoms only? With or without loops? etc).

page 9 line 16/17: "..consensus loop *(Supp. data S1)*."

page 9 line 18: typo catalitic -> catalytic

page 10 line 31: "...unambiguous density for GDP in the A and F monomers...": I assume that there is GTP- -S in the other monomers, correct? It would be nice to show the densities for all nucleotides in a supplemental figure... At least please state for each structure exactly which nucleotides can be clearly observed and which not.

page 11 line 5: Please add the sequence identity between ECMcrC and TgMcrC.

page 11 line 9: typo asp336 -> Asp336

page 17 line 17: "..long-range DNA cleavage..": Please elaborate on why the loss of GTPase activity would abrogate only the long-range DNA cleavage (this is also not covered in the discussion, it probably would be most helpful to include this aspect in a more complete discussion of the final model (with DNA?) towards the end of the manuscript).

page 12 line 6: Is the nucleotide preference of TgMcrB also known? From the structure I would expect TgMcrB to discriminate less against ATP since water molecules are easier to displace, unless there are steric clashes with the amino group of ATP. Please add a sentence if such clashes would be expected from the structure.

page 12 line 10: I would rather say that the "fundamental chemistry" of guanine nucleotide recognition is *different* since different structural elements are used. Please clarify what is meant by "fundamental chemistry".

page 13: I assume that there is no significant sequence identity between the Tg/EcMcrC and the EndoMS (5GKF), please add this information (i.e. state the identity). Please mention if TgMcrC also has a basic patch around the active site like EcMcrC (Nirwan, 2019b). Please add a figure that shows the DNA orientation of the 5GKF complex and the positions of the active sites in the context of the complete TgMcrB^{AAA}C tetradecamer (see Figure B below as example) and discuss that the DNA might be able to bind after a slight opening of the two McrC nuclease domains, but that then the ends of the DNA would run straight into at least one of the hexameric rings, and that the DNA in this position would be perpendicular to the ring plane so that it would be oriented in a direction in which it could thread through the central pore, but that it would be offset by approx. 20Å from the central pore axis, so that it could not pass through the pore in this position (this could be shown as third view in addition to Fig.B below, e.g. in Fig. 6).

Fig B (side and top view of EcMcrBC (6HZ4) superimposed with EndoMS+DNA (5GKF), DNA as gray spheres, McrC in orange, active sites in red spheres:

page 14 line 10: Is the ring opening sufficiently wide to let DNA pass through? It might also be helpful to show a superposition of the closed and "opened" (X-ray) hexamers in top and side views.

page 14 line 23: Please add if it is known that loss of the π -stacking side chain results in loss of the nucleotide affinity (which could explain the structural instability) or if it just results in reduction of the hydrolysis rate.

page 15 line 5: Without being able to look at all structures, the mechanism of GTP hydrolysis activation by McrC described here sounds very similar to the one described in Nirwan, 2019b, except that in their 6HZ4 structure, Lys157 seems not to be involved in the (quite long-distance) hydrogen bonding network leading to the γ -phosphate. Please compare the two mechanisms more closely and also show a figure that illustrates the "repositioning" (line 7) of the asparagines due to McrC compared to the same subunit of the McrB structure without McrC, if possible. Is the density of sufficient quality to allow identification of the side chains?

page 15 line 19 typo: simulated -> stimulated

γ

page 15 line 27: I assume that there is no $\text{PO}_4^{\gamma-}$ in the structure and that the γ -phosphate is from GTP- S. Please clarify.

page 15 line 27-34/page 16 top: Again the description of the "cycling" GTP hydrolysis is very similar to Nirwan, 2019b. Please try to highlight any (potential) differences to their mechanism.

page 16 line 19 and 32: "..basic/fundamental chemistry..": please explain what aspect of the "basic chemistry" is relevant here. I would rather describe it as "..different side chains and secondary structure elements in each structure provide the functional groups that can discriminate for guanine nucleotides" or so.

page 16 line 22: "...domain and or the very start...": Either "and" or "or" is superfluous, or a slash is missing.. please correct.

page 17 line 14: Please mention the smallest diameter of the McrB hexamer without McrC: would the space suffice to accommodate a DNA double helix?

page 18 line 6: I agree that the DNA cannot be cleaved in the position suggested by the superimposition of the current models (see Fig.B). But it looks like that the DNA could be accommodated by a slight rotation of the McrC endonuclease domains so that the DNA would run approximately parallel to the surfaces of the two hexamers in the tetradecameric complex (through the "horseshoe" so to say). Are there any "hinges" in McrC that would allow such a movement?

γ

page 18 line 12: please explain why the structural findings can explain that "GTP- S does not support EcMcrBC DNA cleavage in vitro"

γ

page 18 line 13: Please mention that Pro203 sits in a loop close to the γ -phosphate and the hexamer interface of EcMcrB

Figure 1: please mention the smallest pore diameter (would dsDNA fit through the pore?)

Figure 4 e-j: it would be helpful to add the respective interface names to each panel and also the presumed state (i.e. transition state, post-hydrolysis, π -release, nucleotide exchange, pre-hydrolysis etc).

Figure 6a and b: Please show both figures also in the same orientation and/or add a figure that shows the DNA orientation of the 5GKF complex and the positions of the active sites in the context of the complete TgMcrB^{AAA}C tetradecamer (see example Fig.A), ideally with a (perhaps schematic) indication of the positions of the DNA binding domains, both of TgMcrBC and EcMcrBC. Please explain the "L" in Fig. 6b (linker?). In the legend: Please add "map of the full TgMcrB^{AAA}C complex *at 3.7Å*".

Fig. 6c legend: please add the sequence identity level between TgMcrC and EndoMS (5GKF)

Methods:

Please mention the exact sequences of all constructs, including where the His-tag sits (N-terminal? C-terminal?) and which "artificial" residues (if any) are present after cleavage with the protease.

page 4 line 3: Concentration determination via SDS-PAGE is quite unusual, were there any specific reasons not to choose e.g. extinction measurement?

page 6 line 29 GTPase activity assays at 65°C: There must be quite some spontaneous GTP hydrolysis going on at this temperature, was this taken into account?

page 7 line 17ff: Please state the protein concentrations used for crystallization and the protein buffer.

page 7 line 25 and page 8 line 3: The space group is probably P2₁ according to the suppl. table S1, not P2

page 8 line 2: the information of the native crystal is missing (i.e. crystallization conditions, protein concentration and buffer, cryoprotection)

page 8 line 8: please add the refinement strategy: was NCS used throughout? Constraints? Restraints? TLS refinement? Did TLS refinement lower the R_{free}? What is the percentage of disordered residues? Were there parts with reduced B factors?

page 8 line 19: "..concentration was estimated to be ~14 mg/ml": How was this estimated?

page 9/10: please add angular distribution plots for all structures. Were there any problems with preferred orientations?

page 11 line 18: please add resolution of map (3.7Å)

page 11 line 26: what is the sequence identity of McrC to 1OB8? And the identity between 5GKF and 1OB8? How reliable is the model? How good was the density? How reliable is the sequence assignment in this region?

page 11 line 32: add resolution of the map (2.4Å)

page 12 line 5: What is the sequence identity between TgMcrB and ECMcrB? How reliable is the model and the sequence assignment? Does the sequence assignment correspond 100% to the published structure (e.g. 6HZ4)?

page 12 line 9: Same questions, sequence identity, how reliable is the sequence assignment? Does it correspond 100% to 6HZ4?

Supp. Data S1: please add to legend "Sequence alignment of *the AAA+ domains* of McrB family proteins". Mention that the N-terminal (DNA binding) domains are not conserved.

Supp. Table S2: Perhaps the order of structures would be less confusing if EcMcrB would be the first column, then TgMcrBC, then TgMcrB^{AAA} (to be next to the TgMcrB^{AAA}C structure). Please also add the type of ligand, e.g. 4xGTP- S, 2x GDP etc. for each structure

Response to Referees:

Reviewer #1 (Remarks to the Author):

Despite the fact that McrBC was among the first discovered methyl-directed restriction enzymes, its reaction mechanism remains elusive. Biochemical and EM studies conducted 20-30 years ago suggested that *E. coli* McrBC (EcMcrBC) follows a reaction mechanism reminiscent of Type I restriction enzymes, where NTP hydrolysis powers DNA translocation with concomitant loop extrusion. Recently, Nirwan et al. presented a cryo-EM structure of the EcMcrBC assembly, which consists of two hexameric McrB rings bridged by an McrC dimer.

In the present work, authors present a higher resolution cryo-EM structure of EcMcrBC, and even higher resolution cryo-EM and X-ray structures of McrB subcomplex and McrBC assembly of an McrBC homolog from *Thermococcus gammatolerans* (TgMcrBC). These structures provide new insights into the GTP (vs ATP) specificity of McrBC enzymes and indicate that McrB homo-hexamers have intrinsic asymmetry. Of particular interest is the proposed mechanism of GTP hydrolysis stimulation by McrC, which provides a positively charged "finger domain" residue (R263 in TgMcrBC, K157 in EcMcrBC) to the interface between two McrB subunits, thereby promoting optimal positioning of catalytic residues and ensuring that McrC stimulation at any given time occurs at a single active site.

The manuscript is clearly written, provided results are a valuable addition to the vast field of AAA+ ATPases, and thus this work may be of interest to a broad audience.

We thank the reviewer for the positive assessment of our work.

I have two questions that could be further discussed in this publication:

1. Authors show that the McrB hexamer even in the absence of McrC is asymmetric, with 4 "tight" and 2 "loose" inter-McrB interfaces. But what are the physical constraints of the McrB subunits that result in such obligatory asymmetry? Could it be that the angle between each pair of McrB subunits forming "tight" interfaces is less than 60° (say 59°); to fill the full 360° circle, the angle between the last McrB subunit and adjacent subunits must be greater than 60°, resulting in 2 "loose" interfaces?

What causes the asymmetry in hexameric AAA+ proteins that underlies their function is a fundamental question in the field. In some cases, AAA hexamers are symmetric in the absence of bound substrates, with the processing of substrates inducing biased asymmetry in the hexameric AAA assembly (Huyton et al., 2003, JSB; Bodnar et al., 2018, Nat Struct Mol Biol; Twomey et al., 2019, Science). In the case of TgMcrB, based on the C \$\alpha\$ position of the Lys221 residue in each subunit, the radial sectors occupied by subunits A, B, C, D, E and F are 60°, 59°, 59°, 59°, 59° and 64° (see Figure), indicating that the "wedging" angle of each monomer is likely smaller

than 60°. Therefore, as suspected by the reviewer, the subunits do not perfectly fill the entire circle, resulting in the two loose interfaces at F/A and A/B.

We have updated the first paragraph of the discussion to read:

“In the hexameric arrangement, four of the subunits, B, C, D and E, occupy a radial sector (measured as the radius between the C α positions of Lys221 residues in neighboring subunits) of 59°, with the other two subunits, A and F, occupying radial sectors of 60° and 64°, respectively. This distortion of the hexameric assembly, which results in four tight and two loose interfaces (Figure 1c-e), appears to be maintained by the conformation of key interface residues – Arg360, Glu527 and Tyr530 in one monomer and Arg414, Asp420 and Arg424 in its neighbor – acting *in trans*. Alanine substitutions of these residues increase basal GTPase activity by ~two-fold (Figure S1i). We speculate that interface mutations alter the programmed asymmetry, causing the unrestrained individual subunits to wobble randomly and leading to uncoordinated, stochastic GTP hydrolysis throughout the hexamer.”

2. Based on the proposed role of the McrC "finger" domain in GTP hydrolysis, and the observed interactions between McrB and McrC subunits, authors of the present work rule out the possibility of DNA threading through the central (McrB)₆ pore (authors of the previous cryo-EM study of EcMcrBC, Nirwan et al. 2019, considered such threading of both DNA and McrC through the McrB ring as a possibility). But if DNA is not threaded through the central channels of McrB hexamers, and both McrB rings rotate relative to the (McrC)₂ stalk, it gives us the following mechanism: the two-site DNA is bound by the McrBC tetradecamer, with each site interacting with DNA recognition domain of a different (McrB)₆ hexamer; ensuing rotation of DNA-bound McrB rings relative to McrC would wrap the intervening DNA around the McrBC complex (the dumbbell-shaped McrBC complex would act as a reel); ensuing DNA tension may position it in the catalytic cleft of (McrC)₂, resulting in DNA cleavage. Is such mechanism compatible with available biochemical data?

We agree that – given the structures, the density for the DNA-binding domains in TgMcrB, and the observation that McrC occludes the central pore of the hexamer – such a model is highly tantalizing. Indeed, variations of such a model have been proposed in the past (e.g., Loenen and Raleigh, 2014, NAR). When presented with substrates containing two R^MC-binding sites, EcMcrBC cuts in close proximity to one site or the other and not randomly between them *in vitro* (Panne et al., 1999). Depending on the spacing of the modified sites and how the complex engages them, the arrangement proposed by the Reviewer could be compatible with these observations. However, this reeling model requires that the assembled complex can significantly bend/distort DNA to allow wrapping of the intervening DNA, for which there is currently no supporting evidence. As we note below, to propose a meaningful mechanochemical model for McrBC function, we first need to understand the single-molecule behavior of these complexes, how DNA is engaged by the full assembly, and what other structural changes occur in the process. Since we are currently lacking this information, we have tried to limit our mechanistic interpretations to what is directly supported by our structural and biochemical data.

Minor points:

Fig. S1j - the hexamer opening is indicated for "Homodimer 1" only; it could be shown for "Homodimer 2" as well.

In Figure S1k, we now provide an additional view of the 'open-ring' assembly of the TgMcrB^{AAA} hexamer determined by X-ray crystallography that shows the packing-induced distortion in both

hexamers. In Figure S1j, we now also have an illustration of the crystal packing of TgMcrB^{AAA} in the P2₁ space group to provide additional context for how hexamers interact in the crystal lattice.

page 9 line 18 - "H2Ocatalitic", should be "H2Ocatalytic".

Thank you – we corrected this typo.

Based on the dimerisation mode of McrC, could the DNA stagger (length of 5'- or 3'-overhang) of the cleavage product be predicted?

Superpositions of our structures with the DNA-bound EndoMS dimer (Nakae et al., 2016) suggest that the McrC active sites would contact the two DNA strands apart from each other in a way that would produce cleavage products with sticky ends of ~4-5 bases. The cleavage pattern, however, will depend critically on how the McrC dimer is arranged in its cleavage-competent, active conformation, which is something that is currently unknown. Therefore, to avoid too much speculation that is beyond the topic of our current manuscript, we chose not to discuss the cleavage pattern by the endonuclease domain.

1st paragraph of the "Introduction" and the last paragraph of the "Discussion" look somewhat artificial in the otherwise fluent story.

We value the reviewer's assessment of how these sections may deviate from the detailed, mechanistic insights underlying the bulk of our work here. However, we feel describing how McrBC fits into the broader biological context of anti-phage defense, antibiotic resistance, and phage therapy helps the reader to fully appreciate how this unique molecular machine relates to human health and biology as a whole. This is especially exciting given the recent clinical successes of phage therapy, the ongoing discovery of novel prokaryotic defense systems, and the emergence of different strategies (arn, ipi*, anti-CRISPRs) that phages use to subvert and evade host defenses.

Reviewer #2 (Remarks to the Author):

In this manuscript, Niu et al. Present cryo-EM structures of the McrB hexamer and McrBC complexes from both *Thermococcus gammatolerans* (Tg) and *Escherichia coli* (Ec). McrBC is a two-component modification-dependent restrictions system (MDRS) that restricts phage DNA and foreign DNA containing methylated cytosines. Elucidating the structure and function of bacterial defense systems, such as MDRS, is not only important on a fundamental level, but may aid in the development of new drugs that increase efficacy of phage therapy. While a structure of the AAA+ domain of McrB in complex with McrC was published previously, Niu et al. now provide structures with the full-length Ec and Tg McrB in complex with McrC, and also demonstrate that unbound McrB forms an asymmetric hexamer, which has important implications in the mechanism of MrcB activation by McrC. By comparing structures of Ec and Tg McrBC, the authors show that the two McrBC complexes use different DNA-binding domains that use a general GTP-recognition mechanism employed by all G proteins. Finally, based on the asymmetry present in unbound MrcB and the McrBC complex, and the observed differences in the consensus loop present at the active sites of the McrB hexamer, the authors propose a coordinated and directional GTP-hydrolysis cycle for McrBC.

Major remarks:

1) A previous paper is cited (Nirwan et al., Nature Communications, 2019) that had previously characterized a complex between truncated *E. coli* MrcB and full-length MrcC. While the authors do cite the paper of Nirwan et al., it is not discussed in detail at all, and none of the available structures of the *E. coli* MrcBC complex with three bound GNP and three bound GDP molecules are compared with the GTP γ S and GDP bound structures presented here. While I agree that the paper of Niu et al. provides important new mechanistic insights, I cannot support publication in Nature Communications until the authors explicitly and thoroughly compare their work with the previously published study (see also below), and clearly state the similarities and differences between the structures from the work of Nirwan et al. and the manuscript by Niu et al. presented here.

We agree with the reviewer's point and have updated the text to provide direct comparisons to the previous structure of the truncated *E. coli* MrcBC complex (EcMcrB Δ NC) wherever applicable. This includes new Figure panels, new Supplementary Figures (Figures S5, S9, and S10) and the following specific additions/changes to the Results and Discussion sections:

In the Results section:

Page 7, first paragraph:

"Although the aromatic residue is not strictly conserved across the MrcB family, every homolog contains a residue at this position that is capable of π -stacking (Trp, Phe, Tyr or Arg) (Figure 2c and Supplementary Data S1), including Phe209 in EcMcrB (Nirwan et al. 2019b)."

Page 10, third paragraph:

"The half-complex structure shares the same overall asymmetric architecture as the previously reported structure of the truncated *E. coli* restriction complex that lacks the N-terminal domain of MrcB (EcMcrB Δ NC) (Nirwan et al., 2019b) (Figure S5). Despite being stabilized by different guanine nucleotide analogs (5'-guanylyl imidodiphosphate (GMPPNP) versus GTP γ S), the two models superimpose with an overall RMSD of 2.97 Å, even across the asymmetrically interacting MrcC subunit (Figure S5a). The orientations of interacting subunits are also spatially conserved (Figure S5c-e), suggesting that the assembly and asymmetric architecture of the restriction complex are fundamentally maintained regardless of the used nucleotide analog."

Page 11, third paragraph:

"Although we do not resolve the catalytic or bridging waters in our structure of the *E. coli* complex, the cryo-EM density supports the location of the Lys157 side chain (Figure S6d-f). Lys157 was modeled further away from the signature motif in the EcMcrB Δ NC structure (Nirwan et al., 2019b), possibly owing to weaker density and the lower resolution of the map. The C α positions of this residue and other critical active-site components align with those in our reconstruction (Figure S5b)."

Page 12, second paragraph:

"The backbone carbonyl of Asp176 hydrogen bonds with both the 1' amine and 2' amino group of the guanine base, while the main-chain nitrogen of Phe178 reads out the 6' carbonyl group. The same hydrogen bonds were observed in the truncated, GMPPNP-stabilized EcMcrB Δ NC complex containing residues 162-465 (Nirwan et al., 2019b)."

Page 13, first paragraph:

“EcMcrB Δ NC shows a similar overall arrangement, although numerous single-particle classes with different angles between the two half-complexes were reported for this assembly (Nirwan et al., 2019b). Interestingly, the half-complex reconstruction of the full-length EcMcrBC tetradecamer shows density for an additional ordered EcMcrC nuclease domain (Figure S8a). The organization of the EcMcrC nuclease domains at this dimer interface is identical to that seen in other McrBC complexes (Nirwan et al., 2019b) (Figure 6a and b), with the α 10 helix and an analogous extended loop serving as the primary points of contact (Figure S8b). This observation implies the same tetradecameric assembly is formed by the full-length construct.”

In the Discussion section:

Page 15, second paragraph:

“While all McrBC structures presented here display the same arrangement of four tight interfaces and two loose interfaces, the previous EcMcrB Δ NC structure showed three GMPPNP-bound interfaces and three GDP-bound interfaces in the McrB hexamer, which were likely to be tight and loose interfaces, respectively (Nirwan et al., 2019a). This discrepancy might be due to subtly different binding affinities for nucleotide analogs and the sensitivity of the EcMcrB assembly to nucleotide depletion. TgMcrB, in contrast, exists as stable hexamers even in the absence of any nucleotide (Figure S1a), suggesting that the balance between nucleotide affinity, occupancy, and structural integrity could affect the dynamics of McrB AAA rings.”

Page 15, third paragraph:

“We also note that Trp223 forms a crucial π -stacking interaction with the guanine base that is present in the EcMcrBC reconstructions (Nirwan et al., 2019b) (Figure 5e) and functionally conserved at the sequence level in other homologs.”

Page 17, second paragraph:

“A similar sequential mechanism for GTP hydrolysis and clockwise movement of McrC were previously proposed based on the EcMcrB Δ NC structure (Nirwan et al., 2019b). In the half-complex of the EcMcrB Δ NC tetradecameric structure, three GMPPNP and three GDP were assigned in the subunit interfaces of McrB Δ N (Figure S9v). One of the GMPPNP-bound interfaces (the ‘CD interface’, which corresponds to the E/F interface in this study) was assumed to be the McrC-stimulated active site. We interpret this interface as a post-hydrolysis site and instead believe the stimulation and formation of the catalytic transition state occurs at the adjacent D/E interface. It was further speculated that the interaction of the ‘CD interface’ with the β -sheet ‘stalk’ of McrC initiated GTP hydrolysis (Nirwan et al., 2019b). The resulting conformational changes in the McrB signature motif were not fully appreciated, however, due to the limited resolution of the EcMcrB Δ NC structure. Our cryo-EM reconstruction of TgMcrBC unambiguously reveals the catalytic water molecules and illustrates how McrC’s insertion of a basic residue specifically repositions the signature motif to trigger hydrolysis, providing a chemically and energetically favorable description of stimulated turnover. Given the conserved structural features and asymmetry present in both the Tg and Ec complexes, we anticipate that other McrBC homologs will follow this mechanochemical model.”

Page 19, second paragraph:

“Modeling these interactions in the context of the tetradecameric complexes indicates further steric hindrance: superimposed DNA strands would clash with the McrB subunits (Figures S10c and d, left panels) and have a trajectory that is directed away from the central pore in the hexamer, offset by nearly 30° and 10° for the TgMcrB^{AAA}C and EcmMcrB Δ NC structures, respectively (Figures S10c and d, right panels). These observations support the idea that the current structures represent binding/cleavage-incompatible conformations.”

2) For dataset processed to 2.95 Angstrom with fairly decent statistics, the R/Rfree is excessively high (34.5/36.4). Why is that? Did the authors observe any crystal pathologies such as twinning or anisotropy? Maybe the data could be processed to a lower space group to see if R/Rfree improves and density for the two poorly-resolved molecules improves?

The crystallographic data had no obvious pathologies, including twinning or anisotropy. The high R factor occurs because we cannot properly build/refine the small subdomain of the F subunit, which is highly disordered due to the crystal packing and interactions between adjacent hexamers (See Figure S1j-l). Processing the data in P1 did not improve the density, and the choice of the P2₁ space group yielded the current and best result.

We have added the following sentence to the “Crystallization, X-ray data collection and structure determination of TgMcrB^{AAA}” section of the Supplemental Methods:

“The high R values are due to the crystal packing, in which each hexamer touches a neighboring hexamer, disturbing the position of one of its protomers and causing it to be poorly resolved.”

Minor remarks (all related to the major remark above):

1) In the introduction, the authors cite the paper by Nirwan et al., which provided a first structural view of McrBC by cryo-EM. They state that ‘this work fell short of answering many important mechanistic questions’. It would be crucial if the authors elaborated more on the main findings of the manuscript by Nirwan et al., and on which questions remained unanswered.

We agree with the reviewer’s assessment that our initial description of the previous study was vague.

We have updated the introduction to now read (Page 3, third paragraph):

“A recent cryo-EM reconstruction of the hexameric EcMcrB AAA+ domain bound to EcMcrC at 3.6-Å resolution (Nirwan et al., 2019b) provided the first glimpse of this machine, showing the overall architecture of the complex and proposing a general mechanism for catalytic turnover. However, this study did not resolve the molecular details and chemistry underlying the GTP hydrolysis reaction and its stimulation by McrC, and may not have identified the correct DNA binding mode.”

2) Page 7, lines 5 – 6: The authors write that ‘Trp223, which lies adjacent to the Walker A motif, forms a unique parallel π -stacking interaction from below that has never been observed nor predicted for any GTPase or AAA+ protein’ and that ‘Interestingly, the analogous residue (Phe209) was never mutated in previous studies of EcMcrB as it is not strictly conserved across the McrB family’. Based on what the authors observe in their TgMcrB structure, I find it odd that they do not compare their structure to the previously published one of EcMcrB, which clearly has a π -stacking interaction between Phe209 and the guanine base of GTP γ S (see figure 1, panel g of the paper by Nirwan et al., 2019).

We thank the reviewer for this insight and have amended the text accordingly. We also now cite the manuscript describing the YCJK stress protein (Tsai et al., 2019) that shows a similar π -stacking interaction as noted by Reviewer #3.

In the Results section; Page 6, third paragraph:

“Trp223, which lies adjacent to the Walker A motif, forms an unusual parallel π -stacking interaction from below that is absent in the majority of both GTPases and AAA+ proteins (Iyer et

al., 2004; Leipe et al., 2002), but has recently been observed in the YCJK stress protein (Tsai et al., 2019). Mutation of Trp223 to Ala completely abolishes the basal GTPase activity (Figure 2d) and causes the protein to aggregate, as seen by negative-stain EM imaging (Figure 2e). These observations indicate that π -stacking is critical for both McrB GTP binding and the stability of the oligomeric assembly. Although the aromatic residue is not strictly conserved across the McrB family, every homolog contains a residue at this position that is capable of π -stacking (Trp, Phe, Tyr or Arg) (Figure 2c and Supplementary Data S1), including Phe209 in EcMcrB (Nirwan et al. 2019b).”

In the Discussion section; Page 15, third paragraph:

“We also note that Trp223 forms a crucial π -stacking interaction with the guanine base that is present in the EcMcrBC reconstructions...”

3) Page 10, lines 20-21: The authors start the paragraph with ‘To establish whether different homologs use a conserved mechanism for stimulated hydrolysis, we determined the single-particle cryo-EM structure of the complex formed by the full-length E. coli proteins in the presence of GTP γ S’. I understand that the work on E. coli MrcBC was most probably performed before or during the time the manuscript of Nirwan et al. was published. However, since this happened already in July 2019, the authors cannot pretend this manuscript and the structure described therein do not exist. They should at least compare their structure with the already published one.

As described in our response to the reviewer’s Major remark 1, we have revised the manuscript text to include detailed comparisons to the published EcMcrB Δ NC structure wherever applicable.

4) On page 11 lines 28 - 30 the authors write ‘While we could not calculate a reconstruction for the full EcMcrBC tetradecameric assembly, the map for the half-complex contains density for an additional ordered nuclease domain of EcMcrC (Figure S5f)’. Yet a reconstruction of the EcMcrBAAA+C tetradecameric assembly is readily available (Nirwan et al., 2019). Why is this information not used to compare with the TgMcrBAAA+C tetradecameric structure?

The reviewer raises a valid point. While we did not use their reported structure/model of EcMcrB Δ NC to determine our Ec and Tg structures, the EcMcrB Δ NC structure is a good counterpart to our TgMcrB^{AAA}C structure, because both of them were determined with constructs lacking the N-terminal DNA-binding domains. We therefore added Supplementary Figure S10 to illustrate similarities in pore accessibility and the potential arrangement of DNA relative to the hexamers.

As noted above, we also describe the similarities of these two complexes in the text (Page 13, first paragraph):

“EcMcrB Δ NC shows a similar overall arrangement, although...”

5) As written on page 13, and shown in figure S7, the authors perform modeling of a MrcBC:DNA complex based on a structural comparison with EndoMS. A similar comparison is performed in the paper by Nirwan et al. 2019. (Supplementary figure 13). Similar to what is discussed in Niu et al., Nirwan et al. observe that: ‘a particularly interesting feature of McrB Δ NC is that the McrC stalk blocks the pore of the McrB ring.’ They state that ‘It is possible that the DNA substrate requires sliding in through the widest interface cleft (the FA interface in the structure) or that the enzyme complex disassembles and reassembles on the substrate.’ Why is the latter hypothesis (i.e. complex disassembly and reassembly on the substrate) not discussed by Niu and colleagues?

Although Nirwan et al. propose that the complex could disassemble and reassemble on the substrate, they provide no solid basis or explanation for how this could occur. Our structures – and those described by Nirwan et al. – favor a sequential, rotational model for McrC-stimulated GTP hydrolysis. As we describe in the discussion, this requires the involvement of all subunits, either to be primed, directly interacting with McrC, or freely exchanging nucleotides to maintain a constant directionality. Disassembling the complex and reassembling it on the substrate would break that cycle and could hinder rotational movements if McrC were displaced. It is well established that basal and McrC-stimulated GTP hydrolysis are not required for DNA binding and recognition but play critical roles in cleavage when the R^MC sites are separated by large distances (up to 3 kb). It is thus difficult for us to imagine how translocation could occur in an efficient and energetically favorable manner if substrate engagement would require disassembly and subsequent reassembly of the complex.

It is important to note that Nirwan et al. also raised the idea that DNA could wrap around the external surface of the AAA+ ring without threading through the central channel. Given the lack of experimental evidence supporting either of these differing possibilities and the fact that the DNA-binding domains are poorly resolved in our reconstructions, we purposely stayed away from speculating beyond what we could support with our structural data. Future studies will undoubtedly reveal the mechanisms underlying DNA translocation and cleavage as well as the pathway of DNA during these processes.

Furthermore, Nirwan et al speculate that ‘Like many other AAA+ proteins bound to their DNA substrate, rearrangements at the GDP-bound interface could open the McrB hexamer to form a spiral/lockwasher structure and accommodate the DNA.’ This ‘open spiral’ or ‘lockwasher’ structure of MrcB corresponds to the MrcBAAA+ crystal structure one observed by Niu et al. Could it be that the open spiral state is more than just an artefact imposed by crystal packing?

In the crystal structure of TgMcrBAAA+, one subunit was completely separated from the neighboring subunit by a very large gap, which is not observed in any of the more native-like solution structures we obtained by cryo-EM. Figure S1j now shows how tightly the individual TgMcrBAAA+ hexamers are packed within the crystal lattice, arguing that lattice contacts are the main driving force for the distortions in the crystallized hexamer that are not observed for hexamers in solution. Such large gaps between subunits have been seen in cryo-EM structures of other AAA+ proteins, such as katanin, NSF and the YME1 protease (Punchades et al., 2020, Nat Rev Mol Cell Biol). If the McrBC complex can adopt the open spiral conformation and the gap would accommodate a substrate DNA, it would break the sequential cycle of hydrolysis and prevent McrC from engaging the next active site (effectively inhibiting any potential rotational movements). Large distortions would also likely cause the release of McrC, which occupies the central pore of the hexamer and obscures a clear pathway through which DNA could pass. Figure S10a and b now illustrate the accessibility through this pore and the diameter when McrC is present.

6) In the discussion, the authors propose a sequential mechanism for GTP hydrolysis by McrBC. How does the mechanism compare to the one proposed by Nirwan et al.,?

As noted above, we have added text to the discussion that directly compares our mechanistic model to that of Nirwan et al. (Page 17, second paragraph):

“A similar sequential mechanism for GTP hydrolysis and clockwise movement of McrC were previously proposed based on the EcMcrB Δ NC structure (Nirwan et al., 2019b). In the half-complex of the EcMcrB Δ NC tetradecameric structure, three GMPPNP and three GDP were assigned in the subunit interfaces of McrB Δ N (Figure S9v). One of the GMPPNP-bound interfaces

(the 'CD interface', which corresponds to the E/F interface in this study) was assumed to be the McrC-stimulated active site. We interpret this interface as a post-hydrolysis site and instead believe the stimulation and formation of the catalytic transition state occurs at the adjacent D/E interface. It was further speculated that the interaction of the 'CD interface' with the β -sheet 'stalk' of McrC initiated GTP hydrolysis (Nirwan et al., 2019b). The resulting conformational changes in the McrB signature motif were not fully appreciated, however, due to the limited resolution of the EcMcrB Δ NC structure. Our cryo-EM reconstruction of TgMcrBC unambiguously reveals the catalytic water molecules and illustrates how McrC's insertion of a basic residue specifically repositions the signature motif to trigger hydrolysis, providing a chemically and energetically favorable description of stimulated turnover. Given the conserved structural features and asymmetry present in both the Tg and Ec complexes, we anticipate that other McrBC homologs will follow this mechanochemical model."

Small comments:

Page 3, line 28: 'Stimulation of hydrolysis by a binding partner is also rare among AAA+ proteins'. The E. coli MoxR AAA+ ATPase RavA is actually a nice example of this: binding by the inducible lysine decarboxylase LdcI stimulates RavA activity at low pH (Jessop et al., Commun. Biol., 2020).

We have now included this reference, as well as another reference, Azmi et al., 2008, which describes an additional instance of stimulation (VPS4 by ESCRT-III).

Reviewer #3 (Remarks to the Author):

The manuscript describes cryoEM structures of the AAA+ domain hexamer of the 5-methyl-cytosine-specific restriction enzyme complex B (McrB) from *Thermococcus gammatolerans* with and without the McrC subunit, and of both the *Th.gammatolerans* and *E.coli* structures of the full length Mer BC complexes, in addition to the X-ray structure of the AAA domain hexamer of *Th.gammatolerans* McrB. The structures are very interesting and should be of interest to a wide readership, thus I recommend publication in Nature Communications, following a minor revision to address the points raised in the following.

We appreciate the reviewer's favorable appraisal of our work.

An earlier paper (Nirwan et al, 2019b) also described cryoEM structures of the AAA domain hexamer of *E.coli* McrB in complex with McrC and already found the asymmetric interaction of the stalk of McrC with the distorted McrB-hexamer, which lead the authors to the hypothesis of a consecutive GTP hydrolysis cycle reminiscent of the FI-ATPase. However, the current manuscript adds many structural and analysis details, and the comparison between the *E.coli* and the new *Th.gammatolerans* structures is interesting and merits publication. The presentation is already very nice and easy to follow, but could probably be slightly improved by adding some more comparisons to the previous structure/publication by Nirwan et al (2019b in the current manuscript) and perhaps some additional figures as detailed below.

As this was also suggested by Reviewer #2, we have expanded the text and added additional figures to directly compare our structures to those by Nirwan et al. wherever applicable. For details, please see our response to Reviewer #2's Major remark 1.

The analysis of the X-ray structure is very short, also the description of the corresponding X-ray methods (see also below). The overall temperature factor is very high which probably also

(together with the disordered parts of the structure) explains the quite high R factors. Fig. S 1j (that shows the distorted hexamer) is difficult to interpret since only one orientation is shown, it would be helpful to add e.g. a 90° rotated view. In the given orientation, it does not look like the symmetry-related molecule "pushes" one molecule out of the ring. Is it correct that the ring conformation looks like a snail/spiral? Is the density sufficient to identify the bound nucleotides and/or side chains in the interfaces? Crystal packing forces are usually weak, so the structure might tell something about the mechanism of ring opening/assembly ... Are the GTPase-accelerating side chain interactions that are described for the cryoEM structures still possible in the distorted ring? Would one monomer fall off if the hydrolysis would proceed through the hexamer? Or is opening small enough to keep the interactions? A figure with a superimposition of the Xray/EM rings might be very helpful.

As described in our response to Reviewer #2, the high R factor arises because we cannot properly build/refine the small, highly disordered subdomain of the F subunit. We have now updated Figure S1. Panel S1j shows the tight packing of hexamers in the crystal lattice, panel S1k includes an additional 135°-rotated view (right panel) to show how the symmetry-related molecules interact with each other, and panel S1l shows a comparison between the X-ray and cryo-EM structures of the hexameric rings. The distortion occurs mostly at the subunits that form the loose interfaces (subunits F, A), whereas the subunits that could accommodate the McrC finger domain remained undistorted and retained the bound nucleotides (subunits B, C, D, E). The conformation of the hexamer in the X-ray structure does actually not look like a snail/spiral. Instead subunit A is pushed out from its concentric position and the small domain of neighboring subunit F is disordered.

The idea that the distorted ring might be informative for the mechanism of ring opening/assembly is indeed interesting, but such a discussion would be based on just one observation, which furthermore occurs in the context of an artificial crystallographic environment. As we stated in the discussion "*biochemical characterization of EcMcrBC has shown that DNA binding and GTP hydrolysis are separate and distinct properties in vitro (Gast et al., 1997; Panne et al., 1999; Pieper et al., 1997; Pieper et al., 1999b).*" If the ring distortion/opening were indeed required for DNA "translocation", the GTPase activity would have to be influenced by the presence/absence of the substrate DNA due to the potentially large conformational changes in the complex. In addition, all six McrB subunits interact with McrC at the bottom of the hexamer through the hydrophobic bulky residues in the helix 2 inserts (Figure 3e and Supplementary Data S1). Therefore, neither falling-off of one monomer nor the wide opening of a subunit interface would be likely to occur.

The information about which nucleotide is bound to which interface is apparently incomplete, at least I could not find this for all structures/interfaces. In addition, it would be nice to know how ambiguous/unambiguous the densities for each nucleotide are. Please compare those findings in more detail to Ni1wan et al 2019b who found three GNP and three GDP nucleotides bound to the hexamer; the current manuscript apparently has e.g. five GTP- γ -S and one GDP (is that really GDP?) bound to TgMcrBC (Fig. 4e-j).

We thank the reviewer for this suggestion. We have added new Figure S9 to show the cryo-EM densities for the individual bound nucleotides, Mg²⁺ ions and water molecules. The figure also includes the schematic representation of the subunit numbering and the bound nucleotide species in the structures with comparison to the EcMcrB Δ NC structure. The text has also been updated to include the following statements:

In the Discussion section; Page 16, second paragraph:

“In support of this notion, we find GDP at the A/B site in our EcMcrBC structure (Figure S9a-f and s), the F/A site in the structure of the TgMcrBC complex (Figure S9g-l and t), and at both sites in the structure of the TgMcrBAAAC complex (Figure S9m-r and u). The final tight E/F site likely adopts a post-hydrolysis state that is partially destabilized but still remains intact due to the presence of the γ -phosphate in the bound GTP γ S.”

As stated in our response to Reviewer #2 above, we also describe the possible mechanistic implications of these observations in comparison to the observations of Nirwan et al.

In the Discussion section; Page 17, second paragraph:

“...In the half-complex of the EcMcrB Δ NC tetradecameric structure, three GMPPNP and three GDP were assigned in the subunit interfaces of McrB Δ N (Figure S9v). One of the GMPPNP-bound interfaces (the ‘CD interface’, which corresponds to the E/F interface in this study) was assumed to be the McrC-stimulated active site. We interpret this interface as a post-hydrolysis site and instead believe the stimulation and formation of the catalytic transition state occurs at the adjacent D/E interface....”

Cryo-EM densities for the nucleotides, Mg²⁺ ions, water molecules, and interacting side chains in the McrC-engaged active sites (and the corresponding pocket in the TgMcrB^{AAA} structure) have also been added to Figure S6d-f.

It would of course be extremely interesting to see DNA bound to the tetradecamer. Did you also try to get the DNA complex? If yes, did the double-hexamer fall apart in presence of DNA or was just no DNA bound (similar to the Nirwan structure)? Even a negative stain structure would be very helpful...

We completely agree that a DNA-bound structure would be extremely interesting and informative. Our initial efforts indicate that extensive optimization of the DNA substrate will be required to obtain a suitable sample for structural studies. While we are continuing these efforts, we will not be able to include such a structure in the current manuscript within a reasonable timeframe.

It is evidently very difficult to decide from the current data which of the two main models is correct: The DNA either threads through the pore or binds on the outsides of the hexameric rings. But it seems to me that at least some more conclusions could be drawn from the structures. When I superimposed the Nirwan structures with the EndoMS (5GKF), it seemed like the DNA could be accommodated between the two nuclease domains that bridge the two hexameric rings with relatively modest hinge-bending motions of the McrC moiety as detailed below. (Nirwan 2019b described only the McrC dimer interface as a hinge, is there really only this one hinge possible?) A model with a DNA helix running from the nuclease domains out to the DNA binding domains could then reveal if the base pair distance between the cleavage site and the recognition motif would fit to the biochemical data. It would also be interesting to model the movement of the second hexamer assuming the first one is fixed (by its DNA binding domains) and the McrC stalk would rotate. Could this motion be somehow related to the DNA translocation which also requires GTP hydrolysis?

It is indeed tempting to speculate on the possible arrangements and movements as those described by the reviewer. However, our current understanding of McrBC function is largely based on bulk biochemical experiments and now a defined set of structural snapshots. However, even these new structural snapshots have significant limitations, such as the poorly ordered DNA-binding domains. We feel that proposing a model for the mechanochemical functions of McrBC would be premature at this point and that a meaningful model would require a better

understanding of the conformational changes that occur in McrC, the single-molecule behavior of individual complexes, and how DNA is engaged by the full complex.

The full length structures with the DNA binding domains that are in different positions in EcMcrBC and TgMcrBC (here with a very short linker) are also very interesting, and they would be emphasized by a model that includes the DNA.

We agree that a model with DNA would be informative. However, we feel that such a model would be too speculative at this point given that (1) the DNA-binding domains are poorly ordered and appear somewhat mobile in our reconstructions in the absence of DNA, (2) there is currently no experimental evidence describing potential structural changes in the McrB N-terminus or the McrC nuclease domains that are induced by DNA binding and/or GTP hydrolysis, and (3) the extent to which assembled McrBC complexes can induce bending/deformation/melting of the DNA substrate remains to be defined. We have thus taken a conservative outlook and tried to limit our mechanistic interpretations to what can be deduced directly from our structural results.

minor points (text inside stars is recommended to add to the manuscript):

page 3 line 30: " . .is that *the second component* McrC only exerts ... "

We have revised the text accordingly.

page 4 line 1: " .. A recent cryo-EM reconstitution *at 3.6Å (PDB: 6HZ4, 6HZ7 etc)* of the hexameric ... "

As noted above, we have revised the text to read:

"A recent cryo-EM reconstruction of the hexameric EcMcrB AAA+ domain bound to EcMcrC at 3.6-Å resolution (Nirwan et al., 2019b) provided..."

page 5 line 11-15: Is there any structural explanation for this (surprising) result that the mutations increase the hydrolysis rate?

As noted in our response to a similar question raised by Reviewer #1, this could be related to the "wedging" angle of the individual subunits forming the hexamer. If some subunits assume a "wedging" angle of less than 60°, the remaining subunits would have to cover a larger radial sector to complete the 360° around the ring. This would give rise to tighter interfaces, stabilized by the key residues we describe, and loose interfaces making up the difference. As we note in the text (Page 14, first paragraph of the Discussion section), we speculate that interface mutations alter the programmed asymmetry in the hexamer, causing the unrestrained individual subunits to wobble randomly and leading to uncoordinated, stochastic GTP hydrolysis throughout the hexamer. This would increase the apparent hydrolysis rate in the bulk solution. This observation is consistent with previous results (Piepper et al., 1999a) that mutation of one of the corresponding interface residues greatly attenuated the stimulating effect of EcMcrC on the GTPase activity of EcMcrB, suggesting that stimulation requires the proper asymmetric arrangement of the McrB subunits within the hexamer.

page 5 line 16-24: Description of the X-ray structure is very short, please elaborate, see above. It would also be helpful to add after line 24 that for the solution/modelling of the cryoEM structures only the best defined monomer was used, but no other information from the X-ray structure (if that is correct).

As mentioned above, we have added panels to Figure S1 to describe more clearly the structural deformations observed in the crystal structure and we have modified the description in the Results section to reflect these changes and to provide further detail on which subunits/interfaces are affected (See Page 5, third paragraph).

We also updated the Supplemental Methods describing the model building of the cryo-EM structure as follows (Page 11, third paragraph):

“For the TgMcrB^{AAA} hexamer, the best refined monomer from the X-ray model was used and fit into each subunit density of the 3.1-Å resolution cryo-EM map using UCSF chimera (Pettersen et al., 2004); no other information from the X-ray structure was used.”

page 6 line 24: " .. than the Glu357 Ala mutation *that leads to partial dissociation of the hexamers, probably due to the loss of the magnesium ions and subsequently lower nucleotide affinities* (Figure 2e). "

We feel it would be too speculative to include this statement in the results section as we have not rigorously characterized the structural changes associated with mutations at the nucleotide-binding sites. We prefer to keep the original statement, but we have added additional discussion describing the possible contributions of these residues (see below).

page 6 line 25: What *are* the "different effects on oligomerization .. Asp279 and Glu280 were mutated .. " in EcMcrB (Nirwan, 2019a)? Therein, it was found that D279A/Q prevent oligomerization and do not bind nucleotide, whereas E280A/Q was still hexameric, but catalysis-impaired, which is quite consistent with your observations?

Based on the TgMcrB^{AAA} structure, Asp356 (Asp279 in EcMcrB) may contribute more to the metal coordination for the magnesium ions while Glu357 (Glu280 in EcMcrB) may provide hydrogen bonding to a catalytic water molecule, as is common for the Walker B motif in AAA proteins. This is consistent with the biochemical data presented in Nirwan et al., 2019, as well as with our negative-stain EM data that show less disruption of the hexamer by the Glu357Ala mutation.

We have updated the text accordingly (Page 6, second paragraph):

“This result mirrors the oligomerization defects observed in EcMcrB when the corresponding residues (Asp279 and Glu280) were mutated (Nirwan et al., 2019a), suggesting that the aspartate residue functions in magnesium binding, while the glutamate residue is critical for coordinating the catalytic water as in other AAA+ proteins (Erzberger and Berger, 2006).”

page 7 line 1: TT-stacking of guanine base with Trp/Phe " .. has never been observed .. ": It seems to be infrequent indeed, but e.g. in the YCJX (6NZ4) stress protein of *Shewanella oneidensis* (Tsai et al, 2019) there is Trp10 stacking to the base (see Fig.A below). I did not check other residues besides Trp or all AAA structures ... Please check those structures and phrase this statement more carefully.

At the reviewer’s suggestion, we have revised the text (Page 6, third paragraph):

“Trp223, which lies adjacent to the Walker A motif, forms an unusual parallel π -stacking interaction from below that is absent in the majority of both GTPases and AAA+ proteins (Iyer et al., 2004; Leipe et al., 2002), but has recently been observed in the YCJK stress protein (Tsai et al., 2019).”

page 8 line 1: Did you use the EcMcrC to help with the sequence assignment? What is the sequence identity between EcMcrC and TgMcrC? And/or was the density sufficiently well defined to allow unambiguous assignment in all parts of the McrC?

All models in our study had been built before the EcMcrBC paper by Nirwan et al. was published. While the nuclease domain in the full-length TgMcrBC complex was not well resolved, it was sufficiently well resolved in the cryo-EM map of the TgMcrB^{AAA}C complex to allow unambiguous assignment except for some side chains and a few solvent-exposed loops. A homology model generated based on the *Saccharolobus solfataricus* Holliday junction resolving enzyme (PDB: 1OB8) guided us in building a model for the very C-terminal region of TgMcrC, as described in the Methods section.

The TgMcrC and EcMcrC sequences are 13% identical and 27% similar across the whole protein.

page 8 line 34: The 0.75Å overall RMSD between the hexamer rings TgMcrB^{AAA} and TgMcrC is *extremely* small, please state exactly for which atoms this value was calculated (backbone atoms only? With or without loops? etc).

The RMSD value was calculated between C α atoms of the aligned sequences that were included in both TgMcrB hexamer models, including residues in loops. The hexameric conformation of TgMcrB^{AAA} alone was indeed identical to that of TgMcrB in the TgMcrBC complex.

We have clarified this in the text (Page 8, third paragraph):

"...the conformation of the TgMcrB hexamer remains largely unchanged in the TgMcrBC complex (overall RMSD of 0.75 Å compared to TgMcrB^{AAA} alone, based on the C α atoms..."

page 9 line 16/17: "... consensus loop *(Supp. data S1)*."

We have revised the text accordingly.

page 9 line 18: typo catalitic -> catalytic

Thank you – we have corrected this typo.

page 10 line 31: "... unambiguous density for GDP in the A and F monomers ... ": I assume that there is GTP- γ -S in the other monomers, correct? It would be nice to show the densities for all nucleotides in a supplemental figure ... At least please state for each structure exactly which nucleotides can be clearly observed and which not.

We have added Figure S9 to show the densities for the nucleotides in our maps.

page 11 line 5: Please add the sequence identity between EcMcrC and TgMcrC.

The TgMcrC and EcMcrC sequences are 13% identical and 27% similar across the whole protein. Across the finger domain alone, the sequences are 20% identical and 40% similar.

We have updated the text to include the sequence identity across the region compared (Page 11, second paragraph):

"The finger domains superimpose with an RMSD of 2.4 Å (sequence identity: 20%), confirming the overall structural conservation between these evolutionarily remote homologs."

page 11 line 9: typo asp336 -> Asp336

Thank you – we have corrected this typo.

page 11 line 17: " .. long-range DNA cleavage .. ": Please elaborate on why the loss of GTPase activity would abrogate only the long-range DNA cleavage (this is also not covered in the discussion, it probably would be most helpful to include this aspect in a more complete discussion of the final model (with DNA?) towards the end of the manuscript).

We apologize for the confusing description of the mutation and its effect. We have updated the text to clarify that we are referring specifically to a context in which McrC-stimulated GTP hydrolysis and subsequent translocation is necessary to facilitate cleavage when recognizing methylated sites that are separated by large distances (up to ~3000 bp away), which is established for the EcMcrBC system.

Page 11, fourth paragraph:

"...substitutions would disrupt the hydrogen-bonding network needed to position the catalytic water, leading to a complete loss of GTPase activity and the abrogation of DNA cleavage when translocation is required to engage complexes bound at distant R^MC sites (Pieper et al., 1997; Pieper et al., 1999a)."

We also note later in the Discussion that DNA binding alone might not be capable of inducing a cleavage-competent conformation and that GTP hydrolysis may also be needed for transient reorganization of McrC monomers (see Page 19, second paragraph: "...It remains to be seen whether...").

page 12 line 6: Is the nucleotide preference of TgMcrB also known? From the structure I would expect TgMcrB to discriminate less against ATP since water molecules are easier to displace, unless there are steric clashes with the amino group of ATP. Please add a sentence if such clashes would be expected from the structure.

A rigorous experimental analysis of the nucleotide-binding preferences of TgMcrB has not yet been done, but is planned for future studies. The structure predicts that no steric clashes would occur with ATP, which may have lower affinity as fewer hydrogen bonds are expected to form and the specificity determination in this instance involves water molecules (see Figure 5f-h).

page 12 line 10: I would rather say that the "fundamental chemistry" of guanine nucleotide recognition is *different* since different structural elements are used. Please clarify what is meant by "fundamental chemistry".

The fundamental chemistry of recognition relies on a specific pattern of hydrogen bonding to distinguish between the nucleotide bases. Although different structural elements are used to form these hydrogen bonds with the guanine bases, the recognition of the 2' position is still especially important in both homologs as well as in other GTPases (which use the canonical G4 motif).

page 13: I assume that there is no significant sequence identity between the Tg/EcMcrC and the EndoMS (5GKF), please add this information (i.e. state the identity). Please mention if TgMcrC also has a basic patch around the active site like EcMcrC (Nirwan, 2019b). Please add a figure that shows the DNA orientation of the 5GKF complex and the positions of the active sites in the context of the complete TgMcrBAAAc tetradecamer (see Figure B below as example) and discuss that the DNA might be able to bind after a slight opening of the two McrC nuclease domains, but

that then the ends of the DNA would run straight into at least one of the hexameric rings, and that the DNA in this position would be perpendicular to the ring plane so that it would be oriented in a direction in which it could thread through the central pore, but that it would be offset by approx. 20Å from the central pore axis, so that it could not pass through the pore in this position (this could be shown as third view in addition to Fig.B below, e.g. in Fig. 6).

Fig B (side and top view of EcMcrBC (6HZ4) superimposed with EndoMS+DNA (5GKF), DNA as gray spheres, McrC in orange, active sites in red spheres:

We have added the identity value in the text (Page 13, second paragraph):

“...the coordinates of the *Thermococcus kodakarensis* EndoMS endonuclease (sequence identity: 12%; PDB: 5GKF; Z-score: 7.9; Nakae et al., 2016)....”

Because our model does not include all side chains in the endonuclease domain due to poor density in various regions, we could not unambiguously assign proper rotamers and an explicit location for each residue. Therefore, it is difficult to calculate reliable surface electrostatic potentials by the Adaptive Poisson-Boltzmann Solver (APBS) for comparison. This is further complicated in the TgMcrBC structure, because here the scaffold domain occludes the space of the active site and thus makes it more difficult to render the surface potential.

Panels c and d have been added to Supplementary Figure S10 to indicate the arrangement of the DNA in the EndoMS complex in the context of both the TgMcrB^{AAA}C complex (S10c) and the EcMcrB Δ NC complex (S10d) as well as the angle of offset for the path of DNA relative to the pore.

The text has also been updated as follows (Page 19, second paragraph):

“Modeling these interactions in the context of the tetradecameric complexes indicates further steric hindrance: superimposed DNA strands would clash with the McrB subunits (Figures S10c and d, left panels) and have a trajectory that is directed away from the central pore of the hexamer, offset by nearly 30° and 10° for the TgMcrB^{AAA}C and EcMcrB Δ NC structures, respectively (Figures S10c and d, right panels). These observations support the idea that the current structures represent binding/cleavage-incompatible conformations.”

page 14 line 10: Is the ring opening sufficiently wide to let DNA pass through? It might also be helpful to show a superposition of the closed and "opened" (X-ray) hexamers in top and side views.

The diameter of the central pore at the narrowest constriction is 8 Å, which is not wide enough to allow DNA to pass through. We have included measurements of the pore diameters and accessibility in Figure S10a and b. We cannot accurately calculate this value for the X-ray crystal structure due to disorder in the “pore loop” region associated with the distorted subunit. As

mentioned above, we have added Figure S11 to show the comparison of the X-ray and cryo-EM structures.

page 14 line 23: Please add if it is known that loss of the TT-stacking side chain results in loss of the nucleotide affinity (which could explain the structural instability) or if it just results in reduction of the hydrolysis rate.

The Trp223Ala mutation destabilizes the TgMcrB hexamer and causes significant aggregation (Figure 2e). However, unlike EcMcrB, TgMcrB assembles into stable hexamers even in the absence of any nucleotide analogs. If the Trp223Ala mutation purely decreased nucleotide affinity, we would expect to still see intact hexamers (as we do for the Asp356Ala substitution, which disrupts magnesium coordination and by extension the ability to bind nucleotide). We believe that this mutation has a more complicated effect that ultimately affects structural stability, nucleotide binding, and hydrolysis.

page 15 line 5: Without being able to look at all structures, the mechanism of GTP hydrolysis activation by McrC described here sounds very similar to the one described in Nirwan, 2019b, except that in their 6HZ4 structure, Lys157 seems not to be involved in the (quite long-distance) hydrogen bonding network leading to the -phosphate. Please compare the two mechanisms more closely and also show a figure that illustrates the "repositioning" (line 7) of the asparagines due to McrC compared to the same subunit of the McrB structure without McrC, if possible. Is the density of sufficient quality to allow identification of the side chains?

To illustrate the repositioning of the asparagine, we have added new panels to Figure S6 that show the orientation and the cryo-EM densities for the side chains at the GTP-binding sites and the NxxD motif in TgMcrB^{AAA} (Figure S6d), TgMcrBC (Figure S6e), and EcMcrBC (Figure S6f).

We have updated our description of the EcMcrBC complex to include a more detailed comparison with the EcMcrB Δ NC structure, especially with regard to the Lys157 side chain. The text now reads (Page 11, third paragraph):

"Although we do not resolve the catalytic or bridging waters in our structure of the *E. coli* complex, the cryo-EM density supports the location of the Lys157 side chain (Figure S6d-f). Lys157 was modeled further away from the signature motif in the EcMcrB Δ NC structure (Nirwan et al., 2019b), possibly owing to weaker density and the lower resolution of the map. The C α positions of this residue and other critical active-site components align with those in our reconstruction (Figure S5b)."

page 15 line 19 typo: simulated -> stimulated

Thank you – we have corrected this typo.

page 15 line 27: I assume that there is no PO4 in the structure and that the -phosphate is from GTP- γ -S. Please clarify.

This is correct. We have updated the text to read (Page 16, second paragraph):

"The final tight E/F site likely adopts a post-hydrolysis state that is partially destabilized but still remains intact due to the presence of the γ -phosphate in the bound GTP γ S"

page 15 line 27-34/page 16 top: Again the description of the "cycling" GTP hydrolysis is very similar to Nirwan, 2019b. Please try to highlight any (potential) differences to their mechanism.

As described above in response to a similar comment by Reviewer #2, we have added a more detailed comparison between the two models in the discussion and, wherever applicable, described more explicitly the similarities and differences between our structures and the EcMcrB Δ NC structure presented in Nirwan et al., 2019.

page 16 line 19 and 32: " .. basic/fundamental chemistry .. ": please explain what aspect of the "basic chemistry" is relevant here. I would rather describe it as " .. different side chains and secondary structure elements in each structure provide the functional groups that can discriminate for guanine nucleotides" or so.

The actual "chemistry of recognition" relies on a specific pattern of hydrogen bonding to distinguish between the nucleotide bases. This readout is used by all GTPases and appears to be also utilized by McrB albeit through different structural elements. In McrB homologs, the functional groups come from non-conserved elements, including highly divergent N-terminal segments. Our point was that we illustrate how the same pattern of hydrogen bonding needed for recognition can be achieved even if the structural elements used are not the same (as they are in all other GTPases).

We have amended the text to read (Page 18, first paragraph):

"The subtle distinctions we observe with regard to nucleotide recognition are therefore significant and provide a blueprint for how divergent homologs can maintain the necessary pattern of hydrogen bonding even in radically different structural contexts."

page 16 line 22: " ... domain and or the very start...": Either "and" or "or" is superfluous, or a slash is missing .. please correct.

We have corrected the text to read (Page 17, third paragraph):

"Interestingly, these pieces lie outside the core AAA+ fold and localize to either the flexible linker that connects to EcMcrB's N-terminal DNA-binding domain or the very start of helix α 1 in TgMcrB (Supplementary Data S1 and S2a, colored in gold)."

page 17 line 14: Please mention the smallest diameter of the McrB hexamer without McrC: would the space suffice to accommodate a DNA double helix?

As mentioned above, the diameter of the central pore at the narrowest constriction in the McrB hexameric ring is 8 Å, which is not wide enough to allow DNA to pass through, and the fingertip of McrC completely occludes the pore at the bottom of the hexamer. We have added Figure S10a and b to show the accessible spaces in the center of the McrB hexamer along with associated pore distance measurements.

The text has been adapted (Page 18, second paragraph):

"The asymmetric association of the finger domain's helical bundle with the D/E/F subunits shrinks the pore diameter at the top of the hexamer to ~10 Å (Figure S3b), while the loop-helix-loop region completely occludes the pore at the bottom of the hexamer (Figures 3a, S3e and S10a and b), which narrows to a diameter of ~8 Å even without McrC."

page 18 line 6: I agree that the DNA cannot be cleaved in the position suggested by the superimposition of the current models (see Fig.B). But it looks like that the DNA could be accommodated by a slight rotation of the McrC endonuclease domains so that the DNA would run approximately parallel to the surfaces of the two hexamers in the tetradecameric complex

(through the "horseshoe" so to say). Are there any "hinges" in McrC that would allow such a movement?

The "slight rotation of the McrC endonuclease domains" could indeed orient putatively bound DNA relatively parallel to the surface of the two hexamers. The linker between the finger and scaffold domains and the linker between the scaffold and nuclease domains could potentially act as hinges allowing flexibility to accommodate DNA.

page 18 line 12: please explain why the structural findings can explain that "GTP- γ -S does not support EcMcrBC DNA cleavage *in vitro*"

As cited in the text, it has been shown that GTP γ S does not support McrBC DNA cleavage *in vitro* (Panne et al., 1999). Our structures were obtained in the presence of GTP γ S and all of them show the complex in a cleavage-incompetent conformation. This implies that nucleotide binding and complex assembly with McrC is not sufficient to facilitate cleavage and that remodeling/reorganization powered by GTP hydrolysis is at least partially required.

page 18 line 13: Please mention that Pro203 sits in a loop close to the γ -phosphate and the hexamer interface of EcMcrB

At the reviewer's suggestion, we have revised the description of Pro203 as follows (Page 19, second paragraph):

"Moreover, mutation of Pro203 to valine in EcMcrB, a residue in a loop close to the γ -phosphate and the hexamer interface, significantly reduces both EcMcrC-stimulated GTP hydrolysis..."

Figure 1: please mention the smallest pore diameter (would dsDNA fit through the pore?)

As mentioned above, we have added the information in the discussion (Page 18, second paragraph):

"...finger domain's helical bundle with the D/E/F subunits shrinks the pore diameter at the top of the hexamer to ~ 10 Å (Figure S3b), while the loop-helix-loop region completely occludes the pore at the bottom of the hexamer (Figures 3a, S3e and S10a and b), which narrows to a diameter of ~ 8 Å even without McrC."

Figure 4 e-j: it would be helpful to add the respective interface names to each panel and also the presumed state (i.e. transition state, post-hydrolysis, TT-release, nucleotide exchange, pre-hydrolysis etc).

We have updated panels 4e-j in Figure 4 to include the interface labels in the bottom right hand corner of each panel (e.g., panel 4e is labeled with "D/E" that is also color coded based on the subunits shown). Though we suggest these different conformations reflect snapshots of different states in the hydrolysis cycle, we hesitate to label these as such here as our assignments are still speculative.

Figure 6a and b: Please show both figures also in the same orientation and/or add a figure that shows the DNA orientation of the 5GKF complex and the positions of the active sites in the context of the complete TgMcrBAAAC tetradecamer (see example Fig.A), ideally with a (perhaps schematic) indication of the positions of the DNA binding domains, both of TgMcrBC and EcMcrBC. Please explain the "L" in Fig. 6b (linker?). In the legend: Please add "map of the full TgBcrB"Mc complex *at 3.7Å *".

At the reviewer's suggestion, we have revised the Figure 6a to show the two views of the ribbon model of the tetradecameric TgMcrB^{AAA}C complex instead of the surface map. Additionally, as mentioned above, we have also added new Figure S10, which shows in panels c and d the tetradecameric complexes of TgMcrB^{AAA}C and EcMcrB Δ NC superimposed with the EndoMS-DNA complex (5GKF) in two orientations similar to Figure 6a. Since the surface map was replaced with the ribbon model, the value of the resolution was not included in the legend. We also updated the legend to Figure 6 to clarify the "L" designation as follows (page 35, description of panel 6b): "The α 12 helix and a loop between the β 10 and β 11 strands are labeled as ' α 12' and 'L', respectively."

Fig. 6c legend: please add the sequence identity level between TgMcrC and EndoMS (5GKF)

We added the sequence identity to the figure legend, which is 12%.

Methods:

Please mention the exact sequences of all constructs, including where the His-tag sits (N-terminal? C-terminal?) and which "artificial" residues (if any) are present after cleavage with the protease.

The residue ranges of all constructs were described in the methods. We have added the detailed information on the exact terminal sequences after protease treatment as below:

Page 3, Cloning, expression and purification of TgMcrB^{AAA} section: "Cleavage by HRV 3C protease leaves a glycine and a proline residue immediately upstream of TgMcrB^{AAA}'s N-terminal methionine."

Page 4, Cloning, expression and purification of TgMcrB section: "Cleavage by HRV 3C protease leaves a glycine and a proline residue immediately upstream of TgMcrB's N-terminal methionine."

Page 4, Cloning, expression and purification of TgMcrC section: "...in which a 6xHis tag was introduced upstream of the N-terminal MBP sequence and an HRV 3C cleavage site replaces that for Factor Xa in the multiple cloning site. Cleavage by HRV 3C protease leaves a glycine and a proline residue immediately upstream of TgMcrC's N-terminal methionine."

Page 5, Cloning, expression and purification of EcMcrB section: "Cleavage by HRV 3C protease leaves a glycine and a proline residue immediately upstream of EcMcrB's N-terminal methionine."

Page 6, Cloning, expression and purification of EcMcrC section: "Cleavage by HRV 3C protease leaves a glycine and a proline residue immediately upstream of EcMcrC's N-terminal methionine."

page 4 line 3: Concentration determination via SDS-PAGE is quite unusual, were there any specific reasons not to choose e.g. extinction measurement?

We have found this methodology to be more practicable and, in some cases, critical for the reproducibility of our crystallization experiments, especially when the amount of trace contaminants varies from preparation to preparation or if there are no tryptophans present in a given protein. Both of these issues can dramatically affect the measured OD280 value and thus the calculated concentration of the protein. The "in gel" method bases the measurement only on the amount of the desired protein band present and thus can normalize the value even in the presence of different contaminants. We therefore use this method routinely in our laboratory for all protein purifications.

page 6 line 29 GTPase activity assays at 65°C: There must be quite some spontaneous GTP hydrolysis going on at this temperature, was this taken into account?

We indeed accounted for the spontaneous GTP hydrolysis during the experiments.

We have added the following description to the Supplementary Methods (Page 7, GTPase activity assays section):

“To account for the spontaneous hydrolysis of GTP at 65°C, a protein-free sample containing GTP and magnesium was incubated in parallel and the measured amount of phosphate released at each time point was subtracted from the corresponding measurements of protein-containing samples.”

page 7 line 17ff: Please state the protein concentrations used for crystallization and the protein buffer.

The methods text in the Supplemental Materials has been updated to include the concentrations used for crystallization and the protein buffer:

Supp. Materials, page 4, Cloning, expression and purification of TgMcrBAAA section:

“... For crystallographic analysis, the protein was concentrated to 40-80 mg/mL.”

Supp. Materials, page 7, Crystallization, X-ray data collection and structure determination of TgMcrB^{AAA} section:

“SeMet TgMcrB^{AAA} was diluted to 16 mg/mL with SEC₁₅₀ Buffer containing 2.5 mM GTPγS and crystallized by sitting drop vapor diffusion...”

page 7 line 25 and page 8 line 3: The space group is probably P2₁ according to the suppl. table S 1, not P2

The correct space group is indeed P2₁. We have corrected this error in the Supplemental Methods text.

page 8 line 2: the information of the native crystal is missing (i.e. crystallization conditions, protein concentration and buffer, cryoprotection)

We have added the missing information.

Supp. Materials, page 8, Crystallization, X-ray data collection and structure determination of TgMcrB^{AAA} section:

“TgMcrB^{AAA} with 2.5 mM GTPγS was crystallized at a concentration of 16 mg/mL by sitting drop vapor diffusion in 0.1 M sodium acetate, pH 6.5, 16.75% (v/v) 2-methyl-2,4-pentanediol with a drop size of 2 μL and a reservoir volume of 650 μL. Crystals were frozen as described above.”

page 8 line 8: please add the refinement strategy: was NCS used throughout? Constraints? Restraints? TLS refinement? Did TLS refinement lower the R_{free}? What is the percentage of disordered residues? Were there parts with reduced B factors?

TLS refinement was not performed, although NCS was enforced. No additional constraints were used during the refinement.

The “Crystallization, X-ray data collection and structure determination of TgMcrB^{AAA}” section of the Methods (Page 8, first paragraph) has been updated to reflect this:

“Further model building and refinement was carried out manually in COOT and PHENIX, respectively (Adams et al., 2010; Emsley et al., 2010). NCS was enforced during the refinement with no additional constraints imposed.”

page 8 line 19: “ .. concentration was estimated to be ~14 mg/ml”: How was this estimated?

In this instance, the concentration of the assembled complex was determined by measuring A280.

The “Cryo-EM sample preparation and data collection” section of the Methods (Page 9, first paragraph) has been updated to reflect this:

“...the final concentration was estimated to be ~14 mg/mL by measuring the absorbance at 280 nm.”

page 9/10: please add angular distribution plots for all structures. Were there any problems with preferred orientations?

Because digitonin was used when the samples were frozen, none of the samples suffered from preferred orientations. Angular distribution plots for each structure can now be found in Supplementary Figures S1d, S2c, S4c, and S7a.

page 11 line 18: please add resolution of map (3.7A)

We have added the resolution.

page 11 line 26: what is the sequence identity of McrC to 10B8? And the identity between SGKF and 10B8? How reliable is the model? How good was the density? How reliable is the sequence assignment in this region?

The TgMcrC nuclease domain and 10B8 are 17% identical and 31% similar. 10B8 and 5GKF are 9% identical and 18% similar. The models used to guide building of the nuclease domain share the same structural fold, and we are thus ultimately able to validate features of the model based on the placement of secondary structure elements and conserved PD-(D/E)XK catalytic residues (see Supplementary Figs. 6c and S8c). The quality of the density for the remainder of the structure was sufficient to build and assign the rest of the structure *de novo*.

page 11 line 32: add resolution of the map (2.4A)

We have added the resolution.

page 12 line 5: What is the sequence identity between TgMcrB and EcMcrB? How reliable is the model and the sequence assignment? Does the sequence assignment correspond 100% to the published structure (e.g. 6HZ4)?

The TgMcrB and EcMcrB sequences are 20% identical and 32% similar across the whole protein. Across the AAA+ domain, they are 24% identical and 35% similar. Given the resolution and the quality of the density maps, the models were reliable in that the structure could be traced to account for the entire sequence in all cases and showed good agreement with the published counterparts. For the alignment between the best matched chains (chain D in the EcMcrB structure and chain B in 6HZ4), the C α alignment is 100% with an overall RMSD of 0.75.

page 12 line 9: Same questions, sequence identity, how reliable is the sequence assignment? Does it correspond 100% to 6HZ4?

The TgMcrC and EcMcrC sequences are 13% identical and 27% similar across the whole protein. Across the finger domain alone they are 20% identical and 40% similar. For the alignment of EcMcrC and 6HZ4, the C α alignment is 89% in agreement with an overall RMSD of 1.3. The lower agreement is due to the deviation at the loop regions in the nuclease domains, though the identity in the finger domains is 100%.

Supp. Data S 1: please add to legend "Sequence alignment of *the AAA+ domains* of McrB family proteins". Mention that the N-terminal (DNA binding) domains are not conserved.

We have revised the legend for Supp. Data Figure S1 in the following way:
The title now reads "Sequence alignment of the AAA domains from McrB family proteins."

We added a sentence at the end stating "The N-terminal DNA-binding domains are not conserved and therefore not shown here."

Supp. Table S2: Perhaps the order of structures would be less confusing if EcMcrB would be the first column, then TgMcrBC, then TgMcrB (to be next to the TgMcrBAAAC structure). Please also add the type of ligand, e.g. 4xGTP- S, 2x GDP etc. for each structure

We have reordered the columns as suggested and specified the numbers and types of ligands and metals in the table.

REVIEWERS' COMMENTS

Reviewer #3 (Remarks to the Author):

All issues have been satisfactorily addressed, the manuscript has greatly improved, especially due to the more detailed comparison with the previously published structure.

Some last minor recommendations:

To attribute the high R factors and especially the "ejected" position of one of the protomers in the hexameric ring **solely** to crystal packing might be somewhat unjustified, I would phrase this more carefully.

Usually the crystal packing forces are weak, and the hexamer must be **prone** to (asymmetrically) open at one point, otherwise it just would not crystallize in this space group. I assume it must be more than a crystal packing artifact, most likely one of the (minor) conformations present in solution just got fixed in the crystal.

It would be helpful if the subunits in Supp. Figure S1I were labeled with the corresponding letters. The colors are not sufficient here.

In addition, some of the answers to the reviewer's questions are only in the "response to referees" file, but not in the text, e.g. the information about the sequence identities of the 1OB8 nuclease domain homology model.

I would recommend to also add those to the manuscript text.

Response to Referees:

Reviewer #3 (Remarks to the Author):

All issues have been satisfactorily addressed, the manuscript has greatly improved, especially due to the more detailed comparison with the previously published structure.

We thank the reviewer for their positive assessment of our work and for their helpful suggestions throughout the review process.

Some last minor recommendations:

To attribute the high R factors and especially the "ejected" position of one of the protomers in the hexameric ring **solely** to crystal packing might be somewhat unjustified, I would phrase this more carefully.

The high R values observed during refinement arise (at least in part) from our inability to account for the position of the small subdomain of the F subunit in our model, owing to poorly resolved density in this region near to where each hexamer interacts with its neighbor. We have modified the text in the Methods (Page 25, first paragraph) to clarify this:

"Poorly resolved electron density surrounding the small subdomain of subunit F – near where each hexamer contacts its neighbor in the crystal lattice – and our inability to properly build and refine this portion of the model both contributed to the high R values."

Usually the crystal packing forces are weak, and the hexamer must be **prone** to (asymmetrically) open at one point, otherwise it just would not crystallize in this space group. I assume it must be more than a crystal packing artifact, most likely one of the (minor) conformations present in solution just got fixed in the crystal.

The loose interfaces created by the intrinsic asymmetry of the hexamer are less stable and thus exhibit a greater potential for flexibility and fluctuations. While it remains to be seen if hexamers can sample a fully open conformation in solution (we see no evidence for this in our class averages), we feel it is safe to assume that the lattice packing can influence the hexamer's overall architecture, either by inducing and/or stabilizing an open conformation. We have updated the text in the following ways:

Page 5, second paragraph:

"The distorted appearance of the crystallographic hexamer suggests a greater flexibility at the loose interfaces, which could more readily be influenced by crystal packing forces."

Page 14, second paragraph:

"The asymmetry in the ring also explains how crystal packing forces could induce and/or sustain the open conformation observed in our TgMcrB^{AAA} X-ray structure (Supplementary Figure 1j-l), as the loose interfaces likely have a greater propensity for flexibility resulting from the fewer stabilizing interactions."

It would be helpful if the subunits in Supp. Figure S1l were labeled with the corresponding letters. The colors are not sufficient here.

We have now labeled the individual subunits of both the X-ray and cryo-EM hexamers in Supplementary Figure S1l with the corresponding letters.

In addition, some of the answers to the reviewer's questions are only in the "response to referees" file, but not in the text, e.g. the information about the sequence identities of the 1OB8 nuclease domain homology model.

I would recommend to also add those to the manuscript text.

We have made the following additions to the main text and supplemental information:

Page 13, last paragraph:

"Attempts to model similar interactions with other structurally related homologs like EcoRV⁵⁴ (PDB: 1AZ0; sequence identity: 8% with TgMcrC and 13% with EndoMS) and the *Sulfolobus solfataricus* Holliday junction endonuclease⁵⁵ (PDB: 1OB8; sequence identity: 17% with TgMcrC and 9% with EndoMS) resulted in substantial clashes between the two McrB hexamers."

Methods, Page 28, Model building and refinement section:

"Therefore, a homology search was performed in I-TASSER⁹⁸⁻¹⁰⁰ for TgMcrC residues 312-458, and a homology model was generated based on the *Saccharolobus solfataricus* Holliday junction resolving enzyme⁵⁵ (PDB: 1OB8; sequence identity: 17%)."

Methods, Page 29, Model building and refinement section:

"For the EcMcrBC complex, SWISS-MODEL101 was used to generate a homology model of EcMcrB based on the TgMcrB^{AAA} structure (sequence identity: 20% across the whole protein, 24% across the AAA+ domain alone), which was manually fit into one subunit in the 3.3-Å resolution cryo-EM map using UCSF Chimera."

Supplementary Materials, Page 6, Supplementary Figure 2 legend: Added the following statement regarding the sequence identities of Tg and EcMcrB:

"The TgMcrB and EcMcrB sequences are 20% identical and 32% similar across the whole protein and 24% identical and 35% similar across the AAA+ domain."

Supplementary Materials, Page 11, Supplementary Figure 5 legend: Added the following statement regarding the sequence identities of Tg and EcMcrC:

"The TgMcrC and EcMcrC sequences are 13% identical and 27% similar across the whole protein and 20% identical and 40% similar across the finger domain alone."